# GOLPH3 and GOLPH3L are broad-spectrum COPI adaptors for sorting into intra-Golgi transport vesicles

Lawrence G. Welch, Sew-Yeu Peak-Chew, Farida Begum, Tim J. Stevens, and Sean Munro

**The fidelity of Golgi glycosylation is, in part, ensured by compartmentalization of enzymes within the stack. The COPI adaptor GOLPH3 has been shown to interact with the cytoplasmic tails of a subset of Golgi enzymes and direct their retention. However, other mechanisms of retention, and other roles for GOLPH3, have been proposed, and a comprehensive characterization of the clientele of GOLPH3 and its paralogue GOLPH3L is lacking. GOLPH3's role is of particular interest as it is frequently amplified in several solid tumor types. Here, we apply two orthogonal proteomic methods to identify GOLPH3+3L clients and find that they act in diverse glycosylation pathways or have other roles in the Golgi. Binding studies, bioinformatics, and a Golgi retention assay show that GOLPH3+3L bind the cytoplasmic tails of their clients through membrane-proximal positively charged residues. Furthermore, deletion of GOLPH3+3L causes multiple defects in glycosylation. Thus, GOLPH3+3L are major COPI adaptors that impinge on most, if not all, of the glycosylation pathways of the Golgi.**

## Introduction

Glycosylation is one of the most widespread and heterogeneous posttranslational modifications that can be attached to a plethora of target substrates including proteins, lipids, and RNA (Schjoldager et al., 2020; Maccioni et al., 2011; Sandhoff and Sandhoff, 2018; Flynn et al., 2021). Glycans can have significant impact on the structure, function, and stability of biomolecules, and as a result, glycosylation plays an influential role in many pathological and physiological processes (Pinho and Reis, 2015; Tran and Ten Hagen, 2013; Vajaria and Patel, 2017; Pascoal et al., 2020; Stowell et al., 2015).

Secreted proteins and membrane proteins that traverse or reside in the secretory pathway are predominantly glycosylated in the ER and Golgi during biogenesis (Moremen et al., 2012). Secretory glycosylation involves the sequential addition of glycan moieties, and this controlled sequence of modification is, in part, dependent on the correct compartmentalization of specific glycosylation enzymes across the ER and the different cisternae of the Golgi stack (Moremen et al., 2012; Schjoldager et al., 2020). There are approximately a dozen different glycan modification pathways that act on *N*-linked glycans, *O*-linked glycans, or glycolipids, often with each reaction requiring a unique enzyme (Schjoldager et al., 2020). As a result, the human genome has >200 genes encoding glycosylation enzymes, many of which are Golgi-resident type II transmembrane (TM) proteins

(Lombard et al., 2014). These Golgi enzymes typically have a short cytoplasmic N terminus, a relatively short TM domain (TMD), and an unstructured stem region that acts as a flexible linker between the lipid bilayer and the lumenal catalytic domain (Tu and Banfield, 2010; Welch and Munro, 2019).

For several glycosylation enzymes, the cytoplasmic tail, TMD, and stem (known as the CTS domain) have been shown to be responsible for targeting to the correct sub-Golgi location (Tu and Banfield, 2010; Welch and Munro, 2019). It is likely that the CTS domains act by directing the incorporation of the enzymes into budding COPI vesicles which then recycle them within the Golgi stack (Welch and Munro, 2019; Lujan and Campelo, 2021; Adolf et al., 2019). COPI vesicles are generated by the heptameric coatomer complex and auxiliary proteins including the small GTPase Arf1 (Dodonova et al., 2017; Gomez-Navarro and Miller, 2016). According to the cisternal maturation model, Golgi cisternae continually progress from the cis-Golgi to the TGN, while Golgi residents are segregated away from anterograde cargo into COPI vesicles that bud from the maturing cisternae (Pantazopoulou and Glick, 2019; Glick and Nakano, 2009). These COPI vesicles serve to retrieve Golgi-resident cargoes and deliver them to their correct cisternal location, against the flow of the maturing cisternae. Although other models for Golgi organization have been proposed, the incorporation of Golgi enzymes into intra-Golgi COPI

MRC Laboratory of Molecular Biology, Cambridge, UK.

Correspondence to Lawrence G. Welch: lwelch@mrc-lmb.cam.ac.uk; Sean Munro: sean@mrc-lmb.cam.ac.uk.

vesicles is increasingly well established (Adolf et al., 2019; Dunlop et al., 2017). However, less well understood are the mechanisms by which the CTS domains of the many different enzymes direct them into budding COPI vesicles. Moreover, it is unclear if, and how, these mechanisms differ between vesicles budding from different parts of the Golgi stack, especially as vesicles for retrograde traffic from the early Golgi to the ER are also formed by the COPI coat. Golgi enzymes vary in their distribution across the Golgi stack, implying that there are distinct sorting signals in their CTS domains that serve to maintain this heterogeneous distribution and thus ensure the fidelity of glycosylation (Lujan and Campelo, 2021; Welch and Munro, 2019; Kornfeld and Kornfeld, 1985).

The membrane thickness model proposes that Golgi residents with relatively short TMDs favor a thinner bilayer in budding COPI vesicles over a thick, sphingolipid/sterol-rich membrane that is formed at the late Golgi and then proceeds to post-Golgi compartments (Bretscher and Munro, 1993; Sharpe et al., 2010; van Galen et al., 2014). Other, complementary, models propose that the cytoplasmic tails of Golgi residents interact with the COPI coat, either directly or indirectly through COPI adaptors. Several cis-Golgi–resident enzymes have been reported to bind directly to the COPI coat through a φ(K/R)XLX(K/R) motif in their cytoplasmic tails (Liu et al., 2018). In addition, the COPI adaptor GOLPH3 and its yeast orthologue Vps74 have been shown to be required for the Golgi retention of a selection of glycosyltransferases, and in some cases have been found to bind directly to their cytoplasmic tails (Tu et al., 2008; Schmitz et al., 2008; Isaji et al., 2014; Ali et al., 2012; Chang et al., 2013; Pereira et al., 2014). GOLPH3/Vps74 has been proposed to be recruited to the TGN through an interaction with phosphatidylinositol-4-phosphate (PtdIns4P; Dippold et al., 2009; Wood et al., 2009). Once on the membrane, GOLPH3/Vps74 can simultaneously interact with the COPI coat and sample the tails of the enzyme cargo to package them into vesicles recycling from the TGN to the medial Golgi (Tu et al., 2008; Schmitz et al., 2008; Eckert et al., 2014). Deletion or depletion of GOLPH3/Vps74 can cause the mislocalization of its clients to the lysosome or vacuole for degradation (Schmitz et al., 2008; Tu et al., 2008; Rizzo et al., 2021).

While there seems to be good evidence that GOLPH3 can direct particular enzymes into COPI vesicles, the scale of its contribution to Golgi enzyme retention is still unclear. For instance, it has been recently proposed that GOLPH3 specifically regulates the retention of enzymes involved in glycosphingolipid synthesis (Rizzo et al., 2021). Moreover, several other roles have been proposed for GOLPH3, including regulating Golgi morphology and forward transport from the TGN, raising the possibility that some of the effects on retention may be indirect (Rahajeng et al., 2019; Dippold et al., 2009). In addition, roles for GOLPH3 have been evoked in mechanistic target of rapamycin (mTOR) signaling and in the response to DNA damage (Farber-Katz et al., 2014; Scott et al., 2009). Finally, GOLPH3 has been found to be frequently amplified in various solid tumor types, and its overexpression is associated with poor prognosis (Sechi et al., 2015, 2020; Rizzo et al., 2017). Resolution of the role of GOLPH3 could thus benefit from a comprehensive characterization of its contribution to Golgi enzyme retention. We have

therefore applied two orthogonal, nonbiased, proteomic analyses to identify clients for GOLPH3, and extended this to GOLPH3L, a paralogue that is expressed at low levels in most tissues but whose function is unclear. By using a combination of in vitro binding studies, bioinformatic analyses, and an in vivo Golgi retention assay, we show that both GOLPH3 and GOLPH3L interact with the short cytoplasmic tails of numerous Golgi residents through membrane-proximal polybasic stretches. Deletion of both GOLPH3 genes triggers instability in their clientele, which leads to global defects in glycosylation. Thus, GOLPH3 and GOLPH3L are major, broad-spectrum, cargo adaptors for COPI-coated intra-Golgi vesicles.

## Results

### GOLPH3 and GOLPH3L bind a diverse array of Golgi-resident proteins and the COPI coat

Initially, we used affinity chromatography to identify interactors of GOLPH3 and GOLPH3L. GST fusions to the N terminus of both proteins were used for chromatography of 293T cell lysate. When compared with GST, both GST-tagged GOLPH3 and GOLPH3L enriched a large number of proteins, including many Golgi-resident glycosylation enzymes (referred to by their gene names for simplicity), and all of the subunits of the COPI coat (Fig. 1 A and Data S1). Immunoblotting confirmed the specific enrichment of GALNT7 and β-COP, a Golgi glycosylation enzyme and a COPI subunit, respectively (Fig. 1 B). Among the interacting Golgi enzymes identified by mass spectrometry, several of the previously reported cargo interactors were identified as hits, including GCNT1, EXT1, EXT2, GALNT12, POMGNT1, ST3GAL4, and B4GALT5 (Rizzo et al., 2021; Pereira et al., 2014; Eckert et al., 2014; Isaji et al., 2014; Chang et al., 2013; Ali et al., 2012). Comparing the 73 Golgi-resident membrane proteins enriched by either GOLPH3 or GOLPH3L (P < 0.05 compared with GST alone), there was a high degree of overlap (42 common hits, 13 GOLPH3-specific, and 18 GOLPH3L-specific). In total, 692 proteins were enriched by either GOLPH3 or GOLPH3L (Data S2), with a large proportion being proteins from non-Golgi organelles, which suggests that there is also considerable non-specific binding. Comparing this GOLPH3+3L interactome to a previously reported COPI proteome generated from HeLa cells revealed that of the 249 proteins of the COPI proteome, 102 proteins (41.0%) were also isolated from cell lysate by GOLPH3+3L (Data S2; Adolf et al., 2019). Most of these proteins have not been previously reported to bind either GOLPH3 or GOLPH3L, but many are type II Golgi enzymes. The large proportion of the COPI cargo that are GOLPH3+3L interactors suggests that GOLPH3+3L are broad-spectrum adaptors for COPI-coated vesicles.

### The tails of GALNT2 and ST6GAL1 are sufficient for GOLPH3+3L-dependent Golgi retention

To validate some of the putative GOLPH3+3L clients identified by affinity chromatography, a selection were examined in vivo. Since GOLPH3 is a cytosolic protein that binds the cytoplasmic tails of type II membrane proteins, we used a reporter based on the type II plasma membrane protein sucrase-isomaltase (SI)

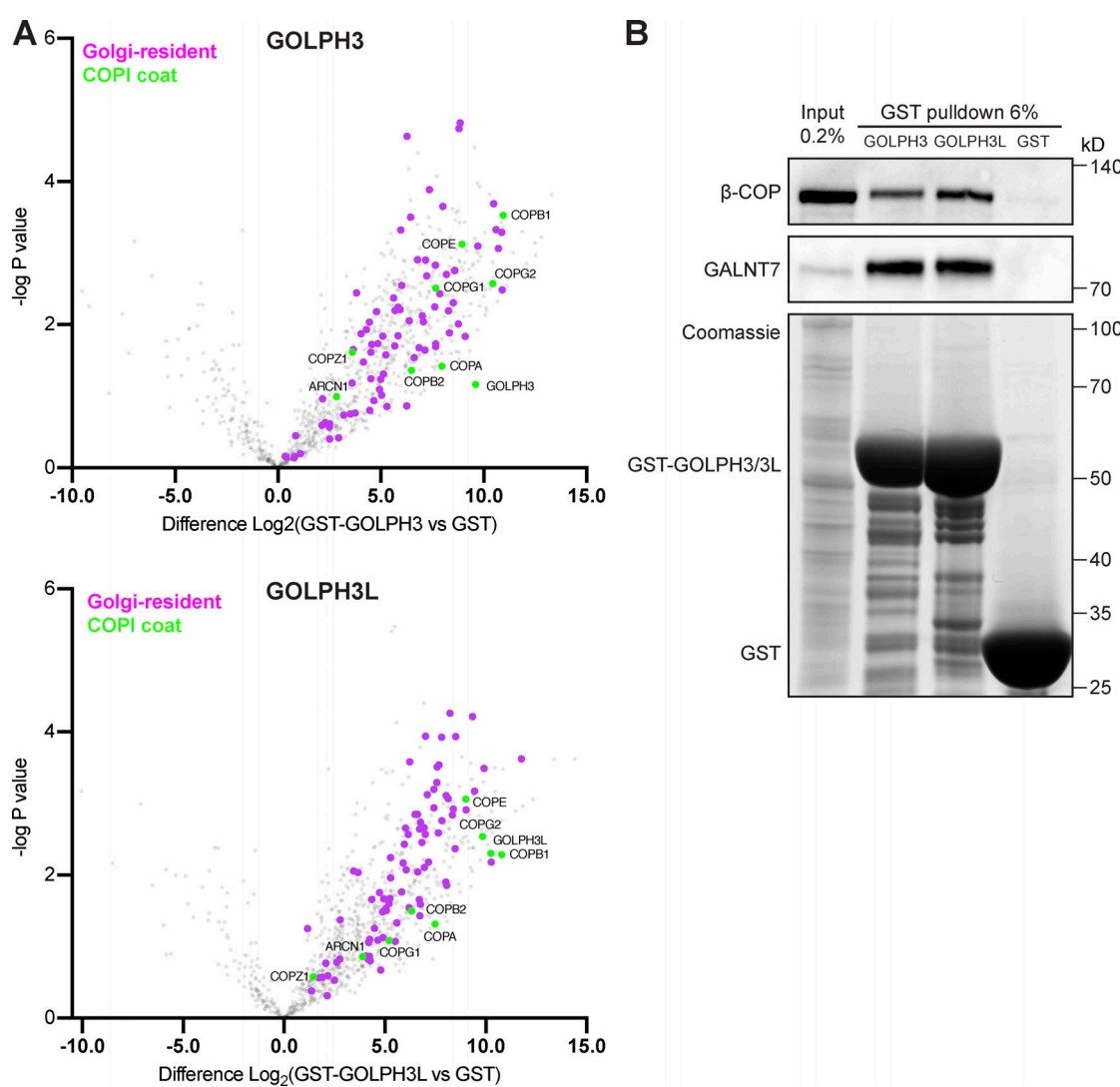

Figure 1. **GOLPH3 and GOLPH3L interact with the COPI coat and a host of Golgi-resident vesicular cargo proteins. (A)** Volcano plots comparing spectral intensity values generated from GST pulldowns from 293T cell lysate using GST-tagged GOLPH3 or GOLPH3L versus GST alone. P values were generated from Welch's *t* tests. Indicated are Golgi-resident integral membrane proteins (magenta, from Swiss-Prot database; see Data S1) and COPI coat subunits (green). Data were from three independent biological replicates analyzed using Perseus. **(B)** Immunoblot of GST pull-downs as in A, β-COP (COPI coat subunit) and GALNT7 (a representative cargo). *n* = 2.

fused to GFP (Fig. 2 A; Liu et al., 2018). We then replaced the cytoplasmic tail of the reporter with the cytoplasmic tails of either a novel GOLPH3 client (GALNT2) or a previously reported one (ST6GAL1; Isaji et al., 2014; Eckert et al., 2014). These reporters were stably integrated into WT U2OS cells or those from which both *GOLPH3* and *GOLPH3L* had been deleted by CRISPR-Cas9 gene editing (Fig. 2 B and Fig. S1). As expected, the SI reporter displayed robust cell surface localization in both WT and Δ*GOLPH3*;Δ*GOLPH3L* U2OS cells (Fig. 2 C). In contrast, the ST6GAL1 and GALNT2 cytoplasmic tail chimeras exhibited a strong Golgi localization in WT cells, but in Δ*GOLPH3*;Δ*GOLPH3L* cells, a considerable proportion localized at the plasma membrane in addition to the Golgi. This is consistent with previous reports that ST6GAL1 is a GOLPH3 client and demonstrates that affinity chromatography has identified a novel client in GALNT2 (Eckert et al., 2014; Isaji et al., 2014; Liu et al., 2018).

## A quantitative Golgi retention assay to interrogate tail- and TMD-dependent retention mechanisms

To quantify the phenomenon observed by immunofluorescence, we used a flow cytometry–based assay. The principle of the assay is that GFP-tagged reporters that are retained in the Golgi will not be accessible to an Alexa Fluor 647-–conjugated anti-GFP antibody added externally at 4°C under nonpermeabilizing conditions (Fig. 3 A). Thus, the ratio of the A647 signal (cell surface signal) to the GFP signal (total cell signal) provides a quantitative measure of retention. As a proof of principle, the GALNT2 cytoplasmic tail chimera and SI reporter cell lines were tested. In WT cells, the SI plasma membrane reporter exhibited a linear relationship between cell surface and total cell signals, with a high ratio between the two, indicative of efficient exocytosis to the plasma membrane (Fig. 3 B and Fig. S2). In contrast, the GALNT2 reporter had a low ratio of cell surface to total

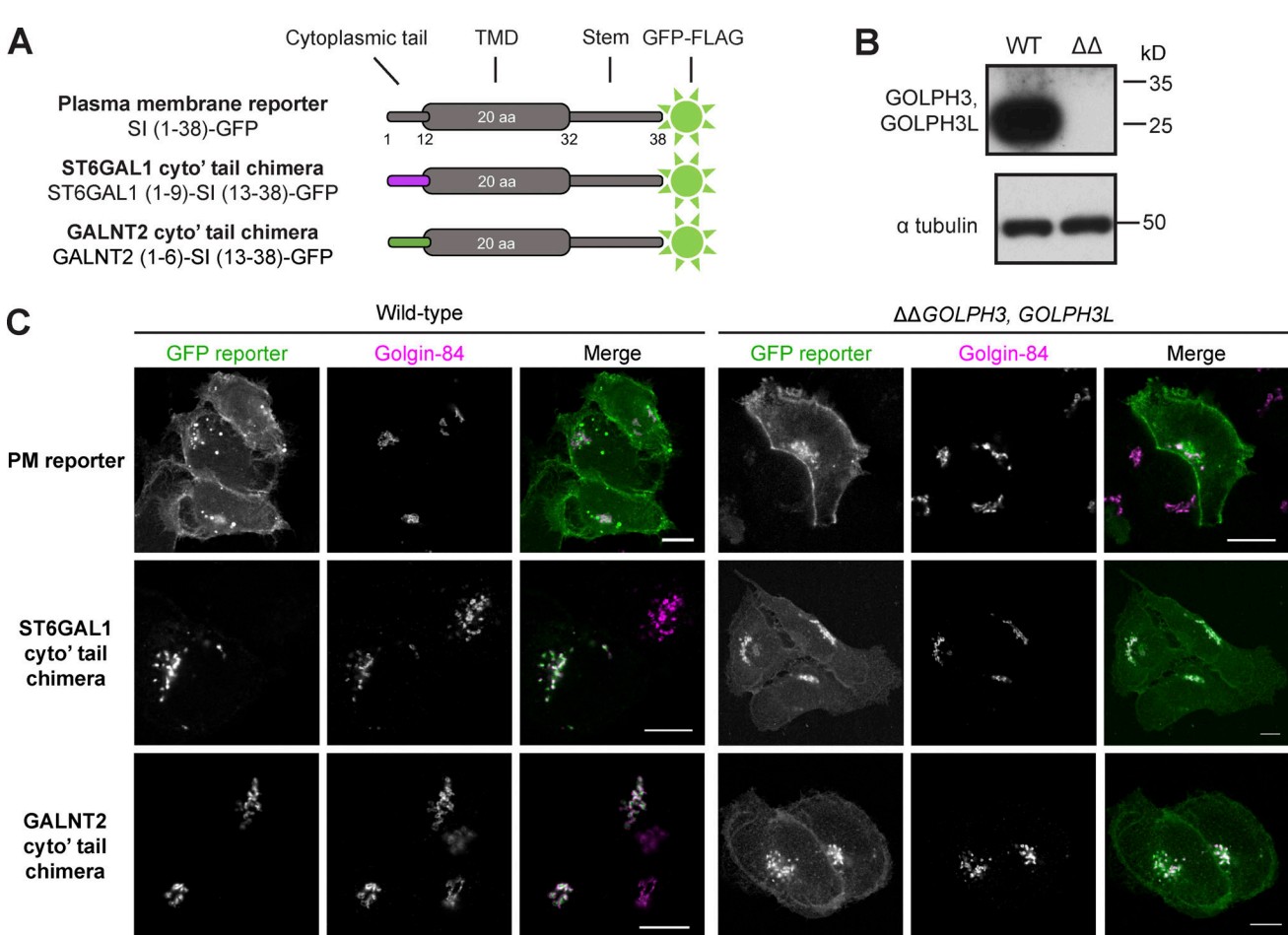

Figure 2. **The cytoplasmic tails of GALNT2 and ST6GAL1 are sufficient to bestow Golgi retention in a GOLPH3+3L-dependent manner. (A)** The GFP-tagged type II TM reporters for Golgi retention. The cytoplasmic tail of the plasma membrane reporter SI was substituted for that of GALNT2 (novel client) or ST6GAL1 (previously reported client). **(B)** Immunoblots of whole-cell lysate from WT U2OS cells and a *ΔΔGOLPH3, GOLPH3L* (ΔΔ) U2OS cell line generated by CRISPR-Cas9 gene editing. **(C)** Confocal micrographs of stable cell lines expressing the indicated GFP-tagged reporters in a WT or GOLPH3 family knockout background. Cells are labeled for golgin-84 as a Golgi marker and a GFP booster for the reporter. The small area of the Golgi makes the intracellular population of reporter clearly visible, and although all the reporters are visible on the plasma membrane in the knockout, the large area and the height of the cell make the surface levels of reporter harder to accurately assess; hence our development of a flow cytometry–based quantitative assay for surface expression. Scale bars, 10 μm.

signal, indicative of Golgi retention. Only at very high levels of expression was the reporter detectable at the surface, indicating saturation of retention. Strikingly, when the GALNT2 reporter was expressed in a *ΔGOLPH3;ΔGOLPH3L* background, it behaved like the SI plasma membrane reporter, confirming that Golgi retention was lost upon the deletion of both GOLPH3 genes.

We next applied this quantitative assay to a wider array of reporters. When the SI plasma membrane reporter was expressed in a *ΔGOLPH3;ΔGOLPH3L* background, its behavior was indistinguishable from that in WT cells: the reporter displayed minimal Golgi retention and robust plasma membrane localization (Fig. 3 C). Although it has been proposed that GOLPH3 is required for efficient anterograde traffic of cargo from the Golgi to the plasma membrane, we could not detect an obvious defect in the traffic of SI upon deletion of *GOLPH3* and *GOLPH3L* (Dippold et al., 2009; Rahajeng et al., 2019). Consistent with the immunofluorescence data, the ST6GAL1 cytoplasmic tail conferred retention in WT cells, and this was mostly relieved in

*ΔGOLPH3;ΔGOLPH3L* cells. We also tested a reporter in which the TMD of SI was replaced with that of ST6GAL1, as its relatively short TMD has previously been shown to be sufficient for Golgi targeting (Munro, 1991; Sun et al., 2021 *Preprint*). The ST6GAL1 TMD chimera also exhibited robust Golgi retention but this was independent of GOLPH3+3L, consistent with the model that GOLPH3 proteins specifically recognize the tails, not the TMDs, of their clients. It has also been reported that the tail of GlcNAc-1-phosphotransferase (GNPTAB) can interact directly with the COPI coat, and that the Golgi retention of a GNPTAB cytoplasmic tail chimera is independent of GOLPH3 (Liu et al., 2018). In accordance with these results, the GNPTAB tail conferred robust Golgi retention that was unperturbed by the deletion of *GOLPH3* and *GOLPH3L* (Fig. 3 D).

### A wide range of Golgi-resident proteins are destabilized by the deletion of GOLPH3 genes

Knockdown of GOLPH3 in mammalian cells, or the deletion of its orthologue Vps74 in yeast, has been found to cause the

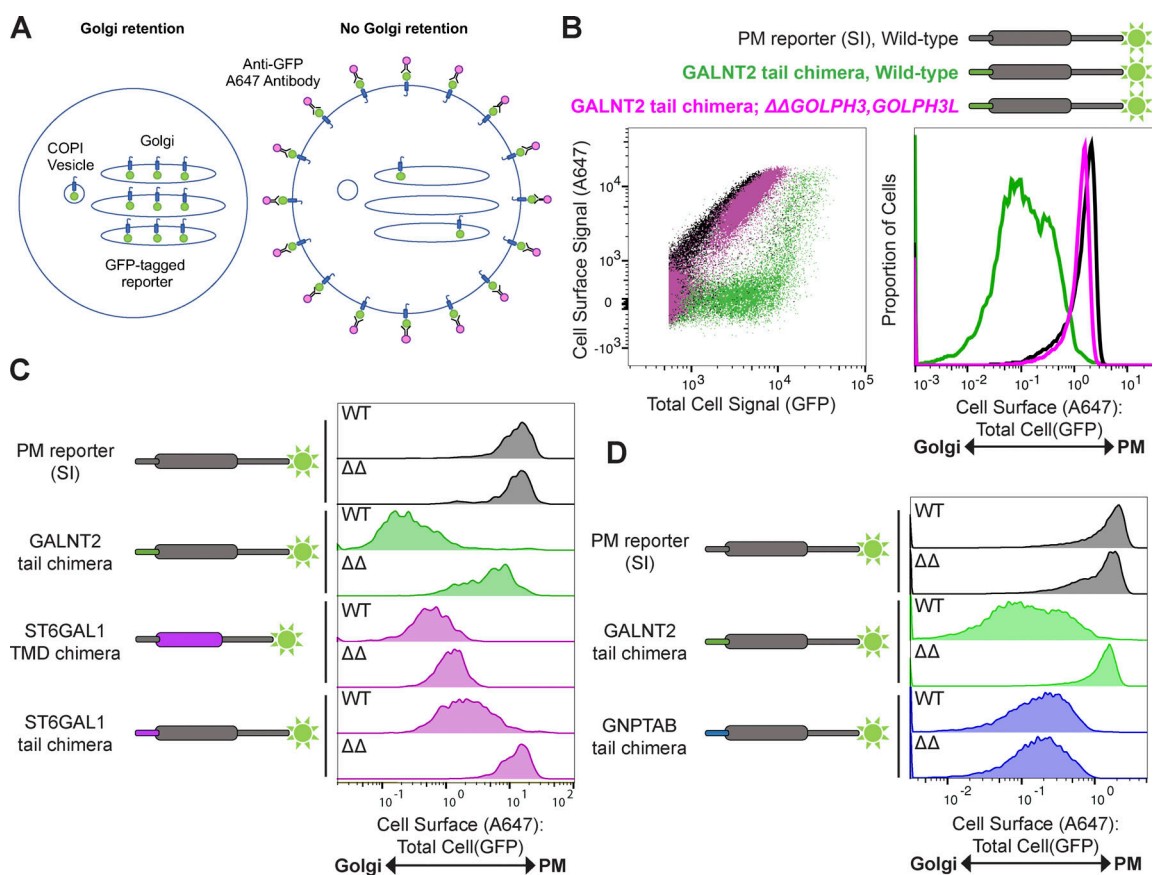

Figure 3. **A quantitative Golgi retention assay to interrogate sorting signals and the contribution of the GOLPH3 proteins. (A)** A schematic of the in vivo Golgi retention assay. Under conditions of Golgi retention, the reporter is sequestered in intracellular compartments (primarily the Golgi and COPI vesicles), and the lumenal GFP-FLAG tag is inaccessible to an A647-conjugated anti-GFP antibody under nonpermeabilizing conditions. In contrast, reporters that are not retained can reach the plasma membrane, where the GFP-FLAG tag becomes accessible to the conjugated antibody. The A647 and GFP signals are then analyzed by flow cytometry. **(B)** Illustrative flow cytometry data for the Golgi retention assay. Overlaid scatter plots (left) of U2OS cells expressing different chimeric reporters in different genetic backgrounds (above) and corresponding histograms displaying the A647:GFP values. Scatter plots and histograms represent 10,000–20,000 events, n = 4. Gating strategy is shown in Fig. S2. **(C and D)** As in B, but histograms represent 500 and 10,000–20,000 events, n = 3 and n = 1, respectively (ΔΔGOLPH3, GOLPH3L [ΔΔ]).

mislocalization of particular Golgi enzymes to the lysosome or vacuole, where they are degraded (Tu et al., 2008; Schmitz et al., 2008; Rizzo et al., 2021; Chang et al., 2013). We therefore tested the effect of removing GOLPH3+3L on the stability of two interactors found by affinity chromatography (GALNT7 and GPP130/GOLIM4) and found that the levels of both were greatly reduced in the double-knockout background (Fig. 4 A). To test the ability of the individual GOLPH3 proteins to rescue this phenotype, each was reintroduced separately using PiggyBac transposition. In both polyclonal populations, the levels of GALNT7 and GOLIM4 were partly restored (Fig. 4 A). Immunofluorescence of the GOLPH3-transduced population revealed considerable heterogeneity in expression levels, suggesting that the partial rescues reflect the presence of low- or nonexpressing cells in the polyclonal populations (Fig. S3 A). For GOLPH3, it was possible to clone individual lines that showed uniform expression and rescue, but this was not possible for GOLPH3L, suggesting that its overexpression may have a dominant negative effect. Nonetheless, when GOLPH3 or GOLPH3L were transiently transfected into the double-knockout cell line, in

both cases cells with a robust rescue of the Golgi accumulation of GALNT7 were clearly present within the population (Fig. S3 B). Thus, the instability of these Golgi residents in the double knockout is a consequence of the loss of the targeted genes, and both GOLPH3 and GOLPH3L can rescue this, and hence confer Golgi retention, individually.

### Proteome-wide analysis of proteins dependent on GOLPH3+GOLPH3L for their stability

The clear effect of removing GOLPH3 and GOLPH3L on the levels of GALNT7 and GOLIM4 suggested that a global analysis of protein levels could complement affinity chromatography as an approach to identifying clients of the GOLPH3 proteins. Thus, we used multiplexed quantitative mass spectrometry based on tandem mass tagging (TMT) to compare WT, ΔGOLPH3; ΔGOLPH3L, and the GOLPH3-rescued cells. This revealed that in the double-knockout cells, many Golgi residents were depleted relative to the WT and rescue cell lines (Fig. 4 B, Fig. S1 D, and Data S3). Moreover, additional proteins, including glycoproteins, also showed changes in abundance. This may, in part,

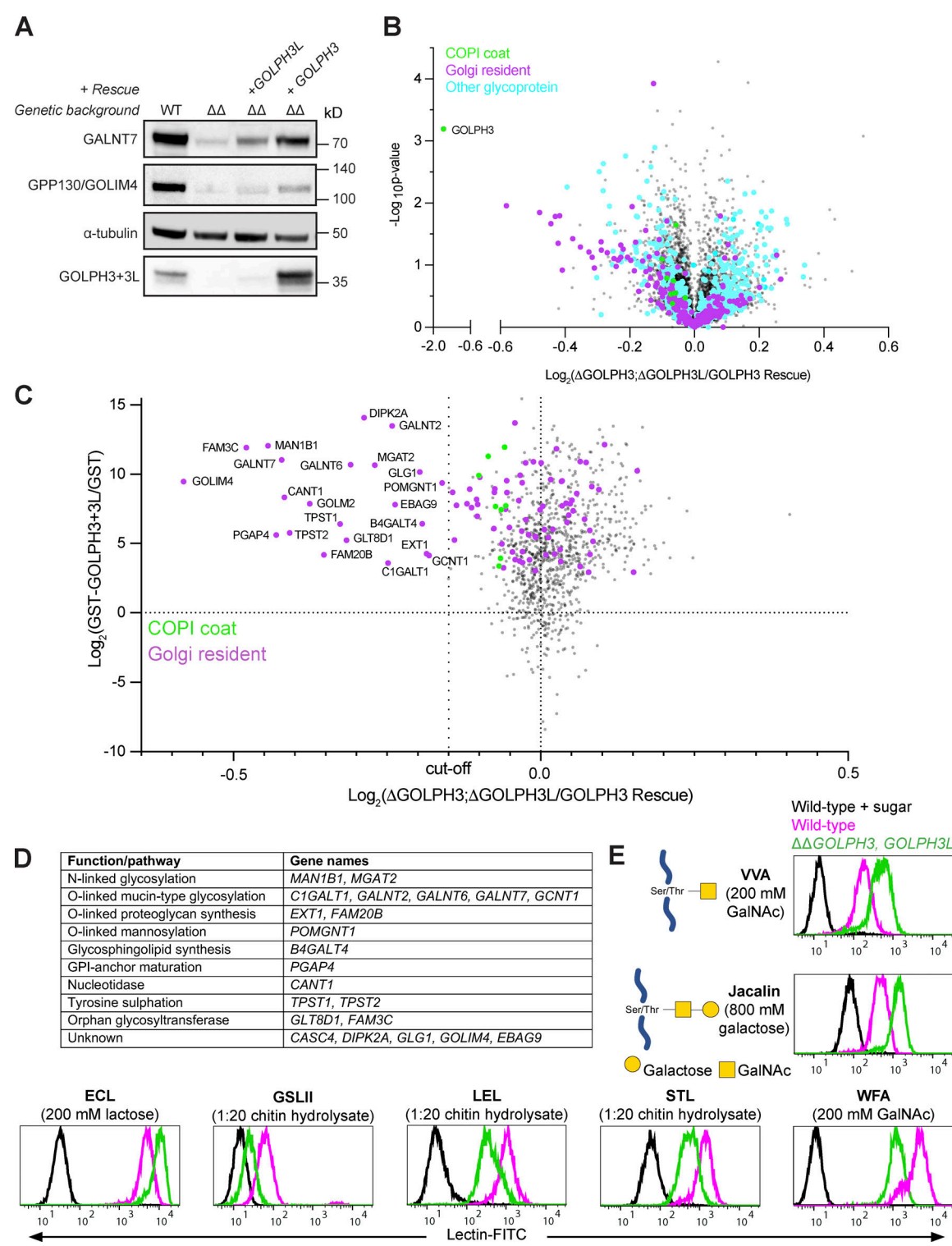

Figure 4. **Deletion of _GOLPH3_ and _GOLPH3L_ results in the destabilization of a diverse array of Golgi-resident enzymes. (A)** Instability of Golgi-resident cargoes (GALNT7 and GPP130) upon deletion of both GOLPH3 genes. Immunoblots of whole-cell lysates from WT, _ΔGOLPH3;ΔGOLPH3L_, and polyclonal rescue U2OS cells. **(B)** Volcano plot comparing spectral intensity values for individual proteins in _ΔGOLPH3;ΔGOLPH3L_ U2OS cells versus _ΔGOLPH3;ΔGOLPH3L +GOLPH3_ polyclonal rescue U2OS cells. The dataset was generated from two repeats and was Z-score normalized according to the median, and P values were generated from a Student's _t_ tests. COPI subunits (green), Golgi-resident integral membrane proteins (magenta), all other glycoproteins (cyan), based on Swiss-Prot (see Data S3). **(C)** Data from A for relative protein abundances in _ΔGOLPH3;ΔGOLPH3L_ cells versus _ΔGOLPH3;ΔGOLPH3L + GOLPH3_ rescue cells plotted against the data for GST-GOLPH3+3L binding versus GST binding from Data S2 (combined dataset in Data S3 C). COPI coat and GOLPH3 proteins (green) and Golgi-resident integral membrane proteins (magenta); the dotted line shows the cutoff for degradation, below which all proteins are Golgi residents. **(D)** A table of the highest confidence GOLPH3+3L interactors. as named in C and defined by showing greater degradation than any non-Golgi protein. All are type II, apart from Glg1 (type I), PGAP4 (three TMDs), and EBAG9 (unclear). **(E)** Flow cytometry of lectin binding to _ΔGOLPH3;ΔGOLPH3L_ and WT U2OS cells. FITC-conjugated

lectins with different specificities (lectins recognizing *O*-linked glycans: VVA and jacalin). Lectin specificity was validated using saturating concentrations of the indicated competing sugar. Histograms are normalized to the mode value for each treatment. At least 10,000 events were collected for each cell line (Fig. S2), and the plots shown are representative of three biological replicates. Symbol nomenclature for glycans was used for illustrations (Varki et al., 2015).

reflect the use of tandem mass spectrometry which increases the sensitivity of protein detection and hence proteome coverage, but comes at the cost of slightly reduced accuracy of quantitation. In addition, it is also known that changes in glycosylation can affect the stability of glycoproteins (Kingsley et al., 1986; Scott and Panin, 2014; Jayaprakash and Surolia, 2017). Thus, to filter for high confidence GOLPH3+3L clients, we compared the data from the proteomic analysis to that obtained with affinity chromatography and found that a set of proteins were strongly depleted in the *ΔGOLPH3;ΔGOLPH3L* cell line and also bound efficiently to GST-GOLPH3+3L (Fig. 4 C). Setting a stringent cutoff based on depletion that was greater than the most depleted non-Golgi protein revealed 22 hits, 17 of which are known Golgi enzymes, with the others being Golgi proteins of unknown function. All but three of the 22 were type II proteins with a single TMD near the N terminus (Fig. 4 C and Data S3). The 17 Golgi enzymes come from a broad array of enzymatic pathways including *N*-linked glycosylation, *O*-linked mucin-type glycosylation, proteoglycan synthesis, *O*-mannosylation, glycosphingolipid synthesis, and Golgi enzymes involved in tyrosine sulphation and nucleotide hydrolysis (Fig. 4 D). Just below the strict cutoff used here were several additional Golgi enzymes, again from a wide range of pathways (Fig. 4 C and Data S3). Thus, combining the two rather noisy datasets reveals a clear set of proteins that are strong candidates to be clients for GOLPH3+3L-dependent Golgi retention, and indicates that GOLPH3+3L act on enzymes from a wide-range of Golgi-localized modification pathways.

The destabilization of a wide range of Golgi glycosylation enzymes should perturb the glycosylation status of the cell surface. To test this, WT and *ΔGOLPH3;ΔGOLPH3L* U2OS cells were probed with a panel of fluorescently labeled lectins that recognize a range of *O*- and *N*-linked glycans (Fig. 4 E). The WT and *ΔGOLPH3;ΔGOLPH3L* cells displayed a marked difference in fluorescence intensity for every lectin tested, consistent with the deletion of *GOLPH3* and *GOLPH3L* causing broad-spectrum defects in glycosylation.

### GOLPH3 recognizes membrane-proximal positively charged stretches

The identification of a set of high-confidence clients for GOLPH3+3L raises the question of what common features they share that allow their recognition. Initially, a biochemical approach was taken to test whether the tails from a range of enzymes were sufficient for binding, thereby excluding the possibility that their retention was indirect by virtue of GOLPH3 binding to an associated protein (McCormick et al., 2000; Hartmann-Fatu et al., 2015; Nilsson et al., 1994). Thus, a series of tail variants of the SI-GFP reporter were generated similar to those used in the Golgi retention assay (Fig. 5 A). The chimeras were overexpressed in 293T cells, and the cell lysate was subjected to affinity chromatography with bacterially expressed GST-GOLPH3. Immunoblots of the eluate revealed that GOLPH3

bound convincingly to chimeras containing tails from several enzymatic pathways including mucin-type *O*-linked glycosylation (GALNT2, GALNT7, and GALNT12), *N*-linked glycosylation (MGAT2 and MANEAL), proteoglycan synthesis (CHSY1, B3GAT3, and EXTL3), tyrosine sulfation (TPST2), sialylation (ST6GAL1), and several orphan proteins (GOLM1, CASC4, and GOLIM4; Fig. 5, B and C; and Fig. S4). A few of the tails showed relatively weak binding to GOLPH3 (GALNT4, MGAT1, and MGAT5) or no detectable binding (FUT3), but none of these showed substantial destabilization with loss of GOLPH3 or were not detected, and so are less likely to be GOLPH3 clients.

A close examination of the tails that bound GOLPH3 failed to reveal an obvious shared sequence motif. However, all tails shared a short length and the presence of clusters of positively charged residues (including the N terminus), with an absence of negatively charged residues. Conversely, tails that displayed poor or no binding to GOLPH3 displayed a relative paucity of positively charged clusters and/or the presence of negatively charged residues. This suggests that GOLPH3 recognizes short, positively charged tails. Moreover, when calculating the predicted net charge of the tails at a cytosolic pH of 7.4, generally only tails with a net charge of ≥4 exhibited robust binding to GOLPH3 in vitro. To test this possible charged-based interaction, mutations were made in the tail of SI in an attempt to bestow GOLPH3 binding. Of all the residues targeted, only mutation of a glutamate to alanine was sufficient to induce GOLPH3 binding, suggesting that negatively charged residues do interfere with GOLPH3 recognition (Fig. 5 C). In addition, basic residues were inserted in the membrane-proximal region of the tail of SI so that the positive charge was increased without the removal of native SI residues. The insertion of three arginines or three lysines was sufficient to bestow binding; however, the insertion of three histidines, which are not expected to be fully protonated at pH 7.4, was not. Furthermore, the binding increased as the number of positively charged residues was increased. When the tails were tested in the in vivo Golgi retention assay, a triple arginine or lysine insertion into the tail of SI was sufficient to confer Golgi retention in a GOLPH3+3L-dependent manner, in accordance with the in vitro findings (Fig. 5, D and E). In summary, GOLPH3 appears to be able to recognize its clients by interacting with their short positively charged tails. This is perhaps best illustrated by the very robust interaction of GOLPH3 with the tail of GALNT2, a tail of only six amino acids, five of which carry a positive charge (MRRRSR). This simple mode of cargo interaction would explain how GOLPH3 can recognize a broad array of clients in vivo.

### Bioinformatic analysis of the GOLPH3+3L clientele

The above results suggest that GOLPH3+3L recognize short, positively charged cytoplasmic tails of type II TM proteins. To see if this correlates with the features of potential GOLPH3+3L clients identified in our two proteomic screens, we applied a

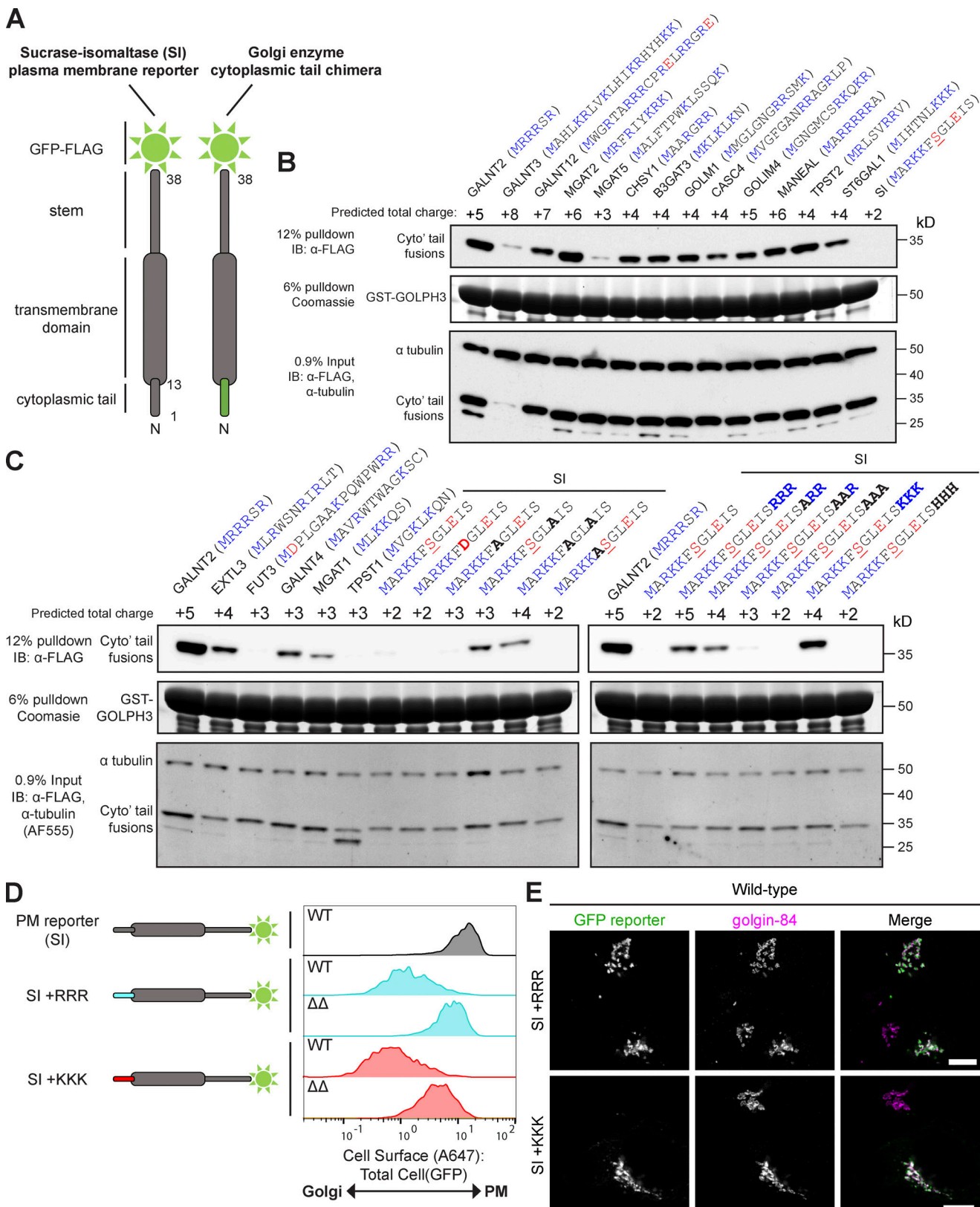

Figure 5. **GOLPH3 interacts with the short, positively charged cytoplasmic tails of a wide range of type II TM proteins of the Golgi. (A)** Schematic showing the Golgi enzyme cytoplasmic tail chimeras used for binding experiments. **(B and C)** Tests of the ability of GST-GOLPH3 to pull down different cytoplasmic tail chimeras from 293T cell lysate. Tail sequences (in brackets) are defined according to Uniprot. Charged residues are colored in blue (positive) or red (negative). The predicted total charges of the tails are based on a cytosolic pH of 7.4 and include the positive charge of the N terminus. The tail of SI is predicted to be phosphorylated at the serine at position 7 (underlined). Bold letters indicate changes resulting from targeted mutagenesis. All blots were

imaged by chemiluminescence unless otherwise stated (Alexa Fluor 555). Data representative of two independent replicates. IB, immunoblotting. **(D)** Histograms displaying the A647:GFP values from a Golgi retention assay comparing U2OS cells expressing the SI reporter with the membrane-proximal insertion of polybasic stretches in different genetic backgrounds (WT or ΔGOLPH3;ΔGOLPH3L [ΔΔ]). Data were collected in the same experiment as Fig. 3 C and thus share the same SI reporter/WT control. See Fig. S2 for gating strategies. Histograms correspond to 500 events and are representative of three independent replicates. **(E)** Confocal micrographs of the indicated GFP-tagged reporters stably expressed in U2OS cells. Cells are labeled with a GFP-booster and for golgin-84 (Golgi marker). Scale bars, 10 μm.

range of bioinformatic analyses. Since the majority of Golgi-resident clients are known to be type II TM proteins, the datasets were filtered for this topology. First, of the type II proteins that were found in the COPI proteome, we compared those that bound to GOLPH3+3L in vitro to those that did not (Fig. 6 A and Data S4). Logo plot analyses showed that those that bind to GOLPH3+3L have a clear enrichment of basic residues next to the TMD, with blank values dominating further from the TMD, indicating that many of the cytoplasmic tails are not longer than 6–10 residues. Leucine residues are the second most abundant in some positions, consistent with reports that an L-x-x-R/K or L-L-R/K-R/K motif contributes to binding to GOLPH3 or its yeast orthologue Vps74 (Ali et al., 2012; Tu et al., 2008; Rizzo et al., 2021). We also applied the same analysis to type II proteins that were classified as degraded or nondegraded in the ΔGOLPH3; ΔGOLPH3L cell line. Among the degraded set, there was a very similar enrichment of membrane-proximal positive residues, again followed by blanks indicative of short cytoplasmic tails.

We also compared the prevalence of short, highly basic, cytoplasmic tails in type II membrane proteins from different subcellular locations. Golgi-resident type II proteins clearly have more membrane-proximal positive charges than ER or plasma membrane residents (Fig. 6 B). Likewise, the type II proteins of the Golgi showed a striking enrichment for shorter cytoplasmic tails compared with those from the ER and plasma membrane (Fig. 6 C). In summary, the proposed GOLPH3+3L-retention signal is greatly overrepresented in Golgi type II TM proteins, further supporting the case that GOLPH3+3L are major COPI adaptors for intra-Golgi vesicles.

Golgi glycosylation enzymes synthesize glycans in a stepwise manner in which each enzyme adds one or more sugars to generate a structure that is then a substrate for the next enzyme in the pathway. We thus examined the position of the GOLPH3+3L clients in the various pathways in which they act. Each enzyme was categorized as early, intermediate, or late-acting based on a recent comprehensive review of glycosylation pathways in human cells (Schjoldager et al., 2020). The glycosylation enzymes that were degraded were almost exclusively early-acting enzymes, whereas the nondegraded group contained more intermediate and late-acting enzymes (26% versus 12%; Fig. 6 D). However, late-acting enzymes were relatively poorly represented across the whole dataset, suggesting that many are of low abundance or absent in U2OS cells. Nonetheless, this suggests that glycosylation enzymes that are degraded upon deletion of GOLPH3 and GOLPH3L generally act early in their respective glycosylation pathways.

## Discussion

Our global analyses of GOLPH3 and GOLPH3L show that both proteins interact with a wide diversity of Golgi-resident enzymes, and for many of these, removal of GOLPH3 and GOLPH3L results in defects in retention of the enzyme in the Golgi apparatus. Previous studies have shown that removal of GOLPH3 alone from mammalian cell lines results in defects in the retention of particular enzymes, but GOLPH3L is present at only ~10% of the level of GOLPH3 in cultured cell lines, and mRNA sequencing analysis indicates a similarly lower expression across most tissues (Bekker-Jensen et al., 2017, GTEx Consortium, 2020). There have been no reported investigations of its function, apart from a suggestion that it is a negative regulator of GOLPH3 function (Ng et al., 2013). Our in vitro binding and in vivo overexpression studies suggest that GOLPH3L has properties similar to GOLPH3, and so it may make a major contribution to retention in some cell types. However, further work will be needed to resolve its precise role, and our inability to isolate cell lines stably overexpressing the protein does suggest at least some potential negative effect at high levels.

Binding assays with GOLPH3 show that a membrane-proximal cluster of basic residues is sufficient for binding and retention, and this feature, combined with a short cytoplasmic tail, is greatly overrepresented in Golgi-resident proteins. This suggests that GOLPH3 and GOLPH3L contribute to the Golgi localization of a wide range of Golgi residents rather than being specific for one particular enzymatic pathway. Nonetheless, it is also clear that other mechanisms can contribute to Golgi enzyme retention, and our data show that retention via direct binding to the COPI coat, or through the TMD, is independent of GOLPH3+3L. This does not preclude some enzymes having multiple retention signals, as is illustrated by ST6GAL1, whose cytoplasmic tail is sufficient for GOLPH3-dependent retention, but which also has a GOLPH3-independent retention signal in its TMD. Such combinations of retention signals could allow precise tuning of the location of an enzyme within the stack or adjusting of the location between different cell types. They also provide a possible explanation for why the contribution of GOLPH3 to the retention of particular enzymes may have been underestimated, as removal of GOLPH3 would not cause a complete loss of Golgi retention. Indeed, the early studies on Golgi enzyme retention that identified the role of the TMD also noted a contribution from the cytoplasmic tail that was not pursued at the time (Munro, 1991; Nilsson et al., 1991; Burke et al., 1994, 1992).

How might GOLPH3 and GOLPH3L recognize a wide range of different proteins? Our data, and those of others, highlight the importance of a cluster of basic residues near the start of the TMD. The structure of GOLPH3 shows a conserved acidic patch that covers much of a flat surface on one side of the protein (Fig. 6 E). Thus, one possibility is that when bound into the forming COPI-coated vesicle, GOLPH3 is held close the bilayer with the acidic patch positioned so as to capture short basic tails

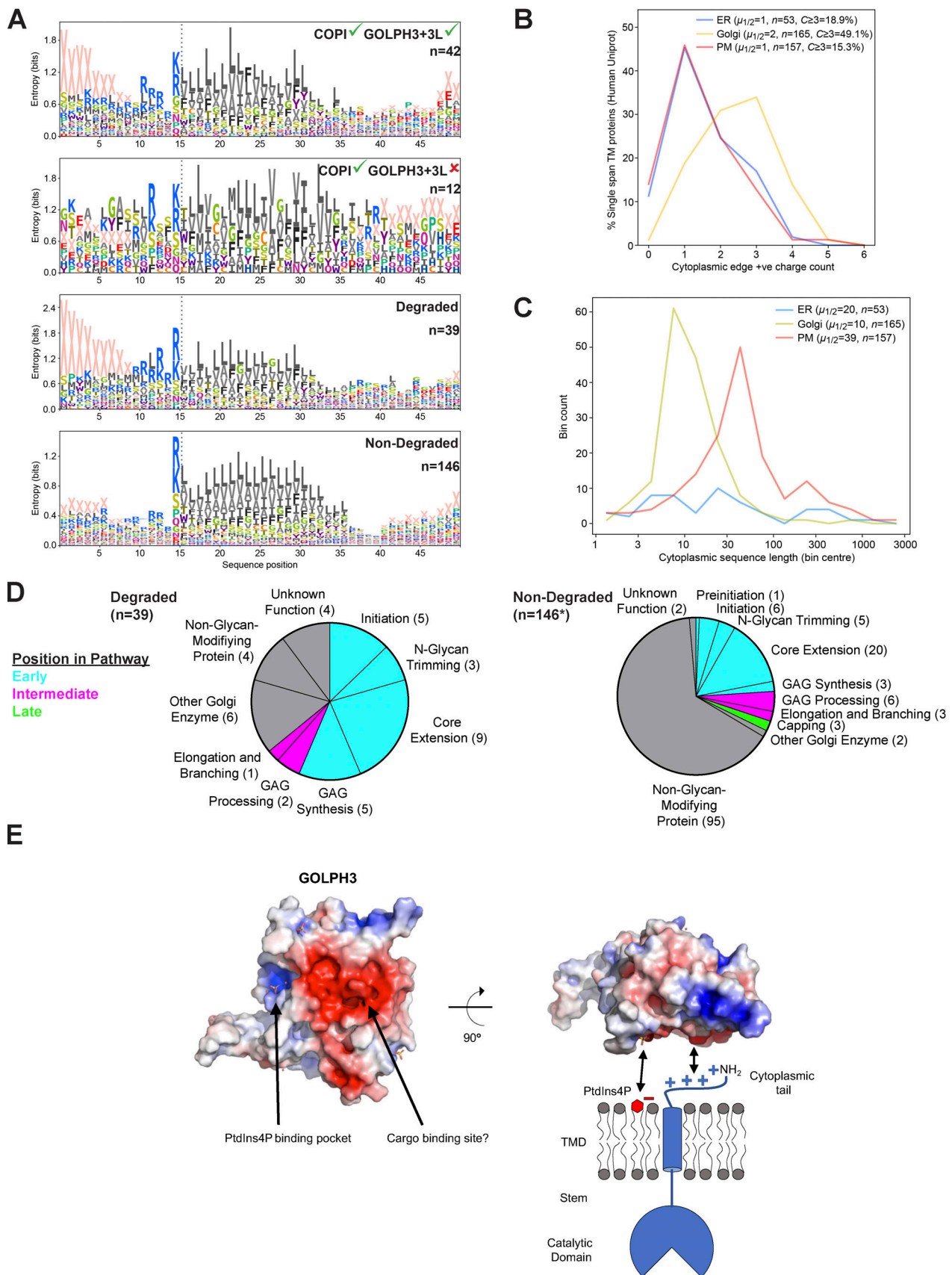

Figure 6. **The clientele of GOLPH3 and GOLPH3L have an enrichment of positively charged residues in the membrane-proximal region of their cytoplasmic tails. (A)** TM span regions of type II membrane proteins from different sets of proteomic analyses, showing differences in relative positional

abundances of amino acids. Sequences were aligned according to the cytoplasmic edge of the TMD (dotted line, with the TMD starting at residue 16). X represents the absence of an amino acid for positions beyond the end of a sequence. The two upper panels are derived from GST-GOLPH3 pulldowns and a reported COPI proteome (see Fig. 5, A–C, proteins with a mean log2 SILAC ratio of >0, sample versus control; in Adolf et al., 2019): of the type II proteins present in the COPI proteome, those enriched in GST pulldowns by GOLPH3+3L were compared with those that were not enriched. The two lower panels are type II proteins that showed the largest reduction in ΔGOLPH3;ΔGOLPH3L cells versus ΔGOLPH3;ΔGOLPH3L + GOLPH3 rescue cells compared with all other type II proteins in the dataset (Fig. 4 B; log2[ΔGOLPH3;ΔGOLPH3L/GOLPH3 rescue] values: degraded less than or equal to –0.1, nondegraded greater than –0.1). Proteins listed in Data S4. **(B)** Quantification of membrane-proximal positively charged residues in the cytoplasmic tails of type II proteins from the ER, Golgi, and plasma membrane (PM; Uniprot). Positive charges were counted within the 6 membrane-proximal residues; arginine and lysine, and the α-amino group for tails ≤6 residues long. **(C)** As in B but comparing cytoplasmic tail length. **(D)** Categorization of degraded and nondegraded proteins according to their position in various glycosylation pathways. Proteins were assigned to different glycosylation pathways according to Schjoldager et al., 2020. *, $n = 155$ rather than 154 since FUT8 can function in both capping and core extension. **(E)** A model for docking of GOLPH3 to membrane and cargo. The structure of GOLPH3 reveals a flat surface containing a large electronegative patch (red, negative; blue, positive; Wood et al., 2009). This electronegative patch could interact with the positively charged residues of the cytoplasmic N termini of clients, and thus recruit them into COPI-coated vesicles.

and exclude proteins with large, folded, cytoplasmic domains that are likely to be destined for the cell surface. Previous studies on particular GOLPH3 clients have suggested that one or more leucine residues in the tail can also contribute to the interaction (Tu et al., 2008; Ali et al., 2012; Welch and Munro, 2019). This feature is clearly not essential, as it is not universal in GOLPH3 clients, but it seems possible that the leucines could bind either at the edge of the acidic patch or back into the lipid bilayer to optimize the interaction of the basic residues with the acidic patch. A proper understanding of the interaction between the tails and GOLPH3 is likely to require structural analysis similar to the cryo-EM studies that have revealed how the COPI coat fits onto bilayers (Dodonova et al., 2017; Bykov et al., 2017). It should be noted that quantitative studies have indicated that GOLPH3 is highly abundant in cultured cell lines, being present at ~50% of the level of the COPI subunits, suggesting that it could contribute to the recruitment of many Golgi residents into a single COPI-coated vesicle.

The COPI coat not only forms the vesicles that bud from Golgi compartments to mediate intra-Golgi traffic, but it also forms the vesicles that capture escaped ER residents in the earliest Golgi compartments to recycle them back to the ER (Bykov et al., 2017; Adolf et al., 2019; Cosson and Letourneur, 1994). Immuno-EM suggests that GOLPH3 is found on the latter half of the Golgi stack, and imaging in yeast indicates that Vps74 is more abundant in the cis and medial cisternae than in the TGN (Wood et al., 2012; Wu et al., 2000). In both yeast and mammals, Golgi recruitment of Vps74/GOLPH3 depends on PtdIns4P, which is relatively enriched at the TGN but apparently present at lower levels on earlier compartments (Graham and Burd, 2011; Wood et al., 2009; Dippold et al., 2009). Indeed, it has been proposed that in yeast the Sac1 PtdIns4P-phosphatase acts in earlier compartments to degrade the PtdIns4P that recycles from later compartments and hence allows the release of Vps74 from membranes; in mammals, disruption of Sac1 activity has been reported to perturb the localization of particular Golgi residents (Wood et al., 2012; Cheong et al., 2010). Further work will be needed to determine the precise distribution of GOLPH3, COPI, and PtdIns4P within the stack, but the dependence on PtdIns4P could allow GOLPH3 to be selectively recruited into the COPI vesicles that form on later cisternae. This would allow COPI vesicles to form at the earliest Golgi compartments without GOLPH3 and hence recycle escaped ER residents rather than Golgi proteins, while the COPI vesicles that form on later

cisternae that contain more PtdIns4P would be equipped with GOLPH3 to mediate intra-Golgi recycling of the Golgi residents.

Previous studies on mammalian GOLPH3 have reported roles for the protein that are distinct from Golgi protein retention—in particular, a role in Golgi morphology and exocytosis mediated by an interaction with myosin-18A, and also roles in DNA repair and mTOR signaling (Farber-Katz et al., 2014; Dippold et al., 2009; Rahajeng et al., 2019; Scott et al., 2009). It is of course possible for one protein to have two or more very different functions. However, our GST-GOLPH3 purification did not reveal binding to either myosin-18A or the retromer complex that was proposed to be responsible for the effects on mTOR. It is possible that the fusion to GST, or the binding conditions used, interfered with these interactions. However, the BioPlex high-throughput analysis of protein interactions obtained similar findings using C-terminally HA-tagged GOLPH3 and GOLPH3L, both of which gave hits with four to five Golgi enzymes, most of which overlapped with our hits, but no hit for either myosin-18A or retromer (Huttlin et al., 2017). In addition, a more recent study of myosin-18A was unable to obtain evidence that it is located on the Golgi (Bruun et al., 2017). Likewise, we were unable to see a difference in the efficiency of cell surface expression of a reporter in the cell line lacking GOLPH3 and GOLPH3L. Consistent with this, a recent study found that Ig is still efficiently secreted by B cells from which Myo18A has been removed (Cheung et al., 2021). Further studies will be required to resolve this issue, especially if some of these putative additional roles of GOLPH3 were to prove to be cell type specific. Nonetheless, we believe that our work, as well as previous studies in mammalian cultured cells and model organisms, provide overwhelming evidence that the major role of GOLPH3 is in Golgi enzyme retention. Interest in the role of GOLPH3 has been increased by the finding that the gene is amplified in a range of solid tumors (Scott et al., 2009; Rizzo et al., 2017; Sechi et al., 2020). A role, or even an exclusive role, for GOLPH3 in retaining enzymes in the Golgi would certainly not be incompatible with these findings, as there is extensive evidence that changes in glycosylation are a hallmark of cancer cells, and they have been linked to increased tumor growth and invasiveness (Stowell et al., 2015).

The retention of resident proteins in the Golgi apparatus has been investigated for three decades, but progress has been complicated by debate over how secretory cargo proteins move through the Golgi stack. Recent studies have provided

near-unequivocal evidence that COPI-coated vesicles selectively recruit Golgi-resident proteins rather than cargo (Adolf et al., 2019). Our findings demonstrate that GOLPH3 does more than contribute to the retention of a few enzymes: it is rather a major adaptor for cargo sorting. Addressing the precise mechanisms by which it binds COPI and its clients, as well as the contribution of TMDs, direct binding, and potentially other adaptors, should provide a clear route to answering the long-standing question of how the Golgi retains its resident proteins as cargo flows past.

## Materials and methods

### Plasmids

Please see Data S5 for a full list of the plasmids used in this study. Sequences encoding GOLPH3 (codon optimized to reduce GC content in the N terminus) and GOLPH3L were synthesized (IDT) and fused at their N terminus to a TEV protease cleavage site and a GST tag in the vector pOPTG (Olga Perisic) for bacterial expression using the restriction sites NdeI and BamHI.

Plasmids were designed for the transient expression of Golgi enzyme chimeric GFP fusions to serve as baits to test in vitro binding to GOLPH3. The short cytoplasmic tail and TMD of SI fused to GFP has been used previously as a type II TM plasma membrane reporter (Liu et al., 2018). An N-terminal section of SI including the tail, TMD and a short lumenal spacer (comparable to the CTS domains of Golgi enzymes) was PCR-amplified from genomic DNA purified from HeLa cells and fused at its C terminus to a GAGA linker, a GFP tag, and a FLAG tag using the restriction sites NheI, KpnI, and NotI in pcDNA3.1+. A host of cytoplasmic tails derived from a variety of Golgi enzymes were used to replace the tail of SI in the plasma membrane reporter. The different tails were introduced into the 3′ end of forward primers, and the chimeras were amplified and cloned into pcDNA3.1+ using the restriction sites NheI and KpnI. Similarly, the TMD of SI was replaced with that of ST6GAL1 through DNA fragment synthesis (IDT) and restriction enzyme cloning using the same restriction sites.

For the purpose of the in vivo Golgi retention assay, a selection of chimeric fusions were subcloned into a modified bicistronic vector used for the generation of puromycin-resistant cumate-inducible stable cell lines using the restriction sites NheI and NotI. The vector was modified from the PiggyBac vector PBQM812A-1 (System Biosciences) from which the IRES (internal ribosome entry site) and downstream GFP were removed. To generate GOLPH3 and GOLPH3L rescue lines, GOLPH3 (kind gift from David Gershlick, University of Cambridge, Cambridge, UK) and GOLPH3L (PCR amplified from the bacterial expression constructs) were inserted upstream of a chimeric intron, an IRES, and mTagBFP2 (synthesized at IDT) by Gibson assembly. Using the restriction sites NheI and NotI, these cassettes were inserted into a modified PiggyBac compatible pcDNA3.1+ vector in which the 5′ and 3′ transposon-specific inverted terminal repeats were inserted upstream of the CMV promoter and downstream of the SV40 poly(A) signal associated with the G418 resistance marker, respectively.

Plasmids were designed to knock out GOLPH3 and GOLPH3L family genes in mammalian cells using CRISPR-Cas9 gene editing. Oligonucleotide pairs encoding single gRNAs (sgRNAs) targeting specific loci were synthesized with overhangs compatible with the restriction enzyme BbsI and annealed together (Data S5). BbsI was used to clone the annealed sgRNAs into the bicistronic CRISPR-Cas9 mammalian expression vector pX458, which encodes Cas9-T2A-GFP under a CAG promoter and a U6 promoter driving expression of the sgRNA.

### Antibodies

Please see Data S5 for a full list of the primary and secondary antibodies used in this study.

### Mammalian cell culture

Human embryonic kidney 293T (ATCC, CRL-3216) and U2OS (ATCC, HTB-96) cells were maintained in a humidified incubator at 37°C with 5% $CO_2$ in culture medium consisting of DMEM (Thermo Fisher Scientific) with 10% FBS (Thermo Fisher Scientific), penicillin-streptomycin, and additional selective antibiotics where specified. Furthermore, stable U2OS cell lines expressing Golgi enzyme chimeric reporters under a cumate-inducible promoter were maintained in the presence of 60 μg/ml cumate (System Biosciences). Cells were passaged every 3–4 d, at which time they were treated with trypsin at 37°C for 2 min, resuspended in culture medium, and diluted by a factor of 1:10. Cells were regularly screened to confirm they were mycoplasma negative using the MycoAlert kit (Lonza).

### Deletion of GOLPH3 and GOLPH3L by CRISPR-Cas9 gene editing

CRISPR-Cas9 technology was used to simultaneously knock out GOLPH3 and GOLPH3L through the induction of frame-shift mutations and the subsequent introduction of premature stop codons in early constitutive exons. GOLPH3 was targeted at one site in exon 2 (target sequence, 5′-GAGAGGAAGGTTACAACT AG-3′) to induce small indel mutations, while two sites 63 bp apart were targeted to introduce a larger out-of-frame deletion mutation in exon 2 of GOLPH3L (target sequence 1, 5′-CTTCTT CCATAAGAGTAAGG-3′; target sequence 2, 5′-GTAATGCAGTTA GGTTTGCT-3′). WT U2OS cells were seeded at a density of 2 × $10^4$ cells/cm² in T-75 flasks in culture medium in a humidified incubator at 37°C with 5% $CO_2$. Once cells were 50–80% confluent, they were transfected with 15 μg of DNA of a bicistronic plasmid encoding the sgRNAs and Cas9-T2A-EGFP. 1 mg/ml polyethylenimine (PEI; Polyscience) in PBS was used for the transfection at a ratio of 3:1 (μL:μg) with DNA in which PEI was incubated in Opti-MEM (Thermo Fisher Scientific) for 5 min before mixing with the DNA. DNA complexes were subsequently incubated for a further 15 min before dropwise addition to cells. 24 h after transfection, GFP-positive cells were sorted to one cell per well into 96-well plates containing fresh culture medium using a MoFlo Cell Sorter (Beckman Coulter), and clones were gradually expanded to 6-well format over the course of several weeks. Whole-cell lysates of clones were analyzed by Western blot to confirm the absence of the protein of interest, and candidate knockout clones were validated by genotyping PCR. Furthermore, the proteome of the final ΔGOLPH3;ΔGOLPH3L U2OS candidate clone was analyzed by mass spectrometry to confirm that the cell line was a true knockout (described below).

## PiggyBac transposon stable cell line generation

The PiggyBac transposon system (System Biosciences) was used to generate stable cell lines expressing either GFP-tagged Golgi enzyme chimeric reporters under a cumate-inducible promoter or *GOLPH3L* and *GOLPH3* under a CMV promoter. WT and Δ*GOLPH3;*Δ*GOLPH3L* U2OS cells were seeded in six-well plates at a density of $2 \times 10^4$ cells/cm$^2$ in culture medium in a humidified incubator at 37°C with 5% $CO_2$. After cells reached 50% confluency, they were transfected with 0.5 µg of a PiggyBac-compatible expression vector and 0.2 µg of PiggyBac transposase (PB210PA-1). 48 h after transfection, cells were expanded to T-75 flasks, and 72 h after transfection, cells were subjected to selection in culture medium with 0.5 µg/ml puromycin (cumate-inducible GFP-tagged reporter cell lines) or 200 µg/ml G418 (GOLPH3 and GOLPH3L rescue cell lines). Cells were cultured under selection for approximately two or three passages to ensure robust selection. Where stated, GOLPH3 and GOLPH3L rescue cell lines were also subjected to cloning by limiting dilution into 96-well plates. Selection was maintained throughout expansion, and the resulting clones were validated by Western blot to select lines with moderate to high expression of the gene of interest where possible.

## GST pulldowns

BL21-CodonPlus(DE3)-RIL competent cells (Agilent) were transformed with constructs encoding GST-GOLPH3, GST-GOLPH3L, or a GST alone, and cells were plated on 2xTY agar plates containing 100 µg/ml ampicillin and 34 µg/ml chloramphenicol and left overnight at 37°C. Single colonies were selected for inoculation of 25-ml overnight liquid cultures of 2xTY containing 100 µg/ml ampicillin and 34 µg/ml chloramphenicol at 37°C at 220 rpm. Starter cultures were used to inoculate larger cultures at a ratio of 1:20, and they were incubated until they reached $OD_{600}$ 0.5–0.8. Cultures were induced with 100 µM IPTG overnight at 16°C. Cells were pelleted by centrifugation at 4,500 $g$ for 15 min at 4°C and washed once with ice-cold PBS by resuspension and centrifugation. Bacterial cells were resuspended in lysis buffer consisting of 50 mM Tris HCl, pH 7.4, 150 mM NaCl, 1 mM EDTA, 0.5% Triton X-100, 1 mM PMSF (Sigma-Aldrich), and 1× cOmplete EDTA-free protease inhibitor cocktail (Roche). Cells were sonicated on ice for 1 min, 10 s on, 10 s off at 45% amplitude using a Sonic Vibra-Cell lance sonicator. Cells were placed on fresh ice for at least 5 min and incubated with agitation at 4°C for a further 10 min. Lysates were clarified by centrifugation at 32,000 $g$ for 10 min at 4°C. Glutathione Sepharose 4B beads (GE Life Sciences) were washed with lysis buffer by resuspension and pelleting by centrifugation at 100 $g$ for 1 min. Clarified bacterial lysates were mixed with the glutathione beads and incubated with agitation at 4°C for 30 min. Beads were subjected to one wash with lysis buffer, one high-salt wash (lysis buffer with 500 mM NaCl), and another four lysis buffer washes. Loaded beads were kept on ice before addition of prey lysates.

Where GST pulldown samples were destined for downstream mass spectrometry analysis, four T-175 flasks of WT 293T cells per bait were grown to confluency. Cells were harvested by resuspension in culture medium, and residual cells were recovered from flasks using an EDTA solution wash. Cell suspensions were pelleted by centrifugation at 300 $g$ for 5 min at 4°C and washed once in ice-cold PBS by resuspension and centrifugation. Cells were resuspended in lysis buffer and sonicated for only 10 s at 45% amplitude using a Sonic Vibra-Cell lance sonicator. Lysates were clarified by centrifugation at 16,100 $g$ for 10 min at 4°C, and the supernatant was mixed with the preloaded beads and incubated with agitation for 1–2 h at 4°C. Beads were washed five times with lysis buffer, and specific interactors were eluted in lysis buffer with 1.5 M NaCl. High-salt elutions were subjected to TCA/acetone reprecipitation and resolubilized in 1× LDS with 10% β-mercaptoethanol (BME). Bait proteins were eluted by boiling in 2× LDS with 10% BME.

For GST pulldown experiments involving GFP-tagged chimeric reporters, 293T cells were seeded in T-75 flasks in culture medium. Once cells reached 50–80% confluency, cells in each flask were transfected with 15 µg of plasmid DNA encoding the chimeric reporters using PEI as described above. 48–72 h after transfection, cells were harvested, washed, and lysed as for samples destined for use in downstream mass spectrometry analysis. GST-GOLPH3–loaded beads were split evenly among the different reactions, mixed with the lysates containing the different chimeric GFP-tagged reporters, and incubated with agitation for 1 h at 4°C. A fraction of the clarified lysate was retained for use as input controls. Beads were washed five times with lysis buffer, and proteins were eluted by boiling in 2× LDS with 10% BME or 50 mM TCEP, pH 7.0.

## Mammalian cell lysis

WT, Δ*GOLPH3;*Δ*GOLPH3L*, and rescue U2OS cell lines were seeded at a density of $2 \times 10^4$ cells/cm$^2$ in six-well plates or 100-mm dishes in the selection medium or culture medium. Once cells reached 80–90% confluency, cells required for Western blotting were washed once with EDTA solution, incubated in trypsin solution for 2 min at 37°C, and resuspended in culture medium. Cells required for whole-cell proteomic analysis by mass spectrometry were washed once with ice-cold PBS and detached from flasks by scraping in PBS. All cell suspensions were pelleted by centrifugation at 300 $g$ for 5 min at 4°C and washed once in ice-cold PBS by resuspension and centrifugation. Cells for Western blotting were resuspended in lysis buffer, while cells required for mass spectrometry were lysed in 8 M urea and 20 mM Tris HCl. All cells were sonicated for 1 min using a Misonix 300 water sonicator for 1 min for 10 s on, 10 s off at amplitude 5.0. Lysates were cleared by centrifugation at 16,100 $g$ for 10 min at 4°C. The protein concentration of the lysates was measured using a Pierce BCA Protein Assay Kit (Thermo Fisher Scientific) and Infinite F200 plate reader (Tecan). The protein samples for Western blotting were normalized across treatments and mixed with loading dye and reducing agent to a final concentration of 1× LDS and 50 mM TCEP pH 7.0. Protein samples for mass spectrometry were diluted to 2 µg/ml, snap frozen in liquid nitrogen, and stored at –80°C until required.

## SDS-PAGE and immunoblotting

Protein samples were incubated at 90°C for 3 min, loaded into Novex 4–20% Tris-Glycine Mini Gels (Thermo Fisher Scientific),

and resolved for 1 h at a constant voltage of 175 V in Tris-glycine SDS running buffer. Total protein in the gels was stained with InstantBlue Coomassie stain (Expedeon) for 1 h to overnight at room temperature with agitation. Gels were washed five times for 5 min in $H_2O$ before imaging. Alternatively, gels were subjected to a Western blot in which protein was transferred onto a 0.45-µm nitrocellulose membrane using a Mini Trans-Blot Cell (Bio-Rad), in transfer buffer in the presence of an ice block for 1 h at a constant current of 255 mA. Blots were blocked in 3% wt/vol nonfat dry milk in PBST (0.1% Tween-20 in PBS) for 1 h at room temperature with agitation. Blots were incubated with the primary antibody diluted in 3% milk in PBST overnight at 4°C with agitation. Blots were washed four times for 5 min in PBST and incubated with the secondary antibody diluted in 3% milk in PBS for 1 h at room temperature with agitation. Blots were washed four times for 5 min in PBST. Where applicable, chemiluminescent substrates (Amersham ECL or Amersham ECL Prime, Cytiva) were added to blots 3 min before exposure to x-ray films, which were developed using a JP-33 film processer (JPI Healthcare Solutions). Alternatively, blots were imaged using a ChemiDoc (Bio-Rad). Where specified, blots stained with an AF555-conjugated secondary antibody were also visualized using a ChemiDoc.

## Lectin binding

WT and ΔGOLPH3;ΔGOLPH3L U2OS cell lines were seeded at a density of $2 \times 10^4$ cells/cm² in T-75 flasks in culture medium in a humidified incubator at 37°C with 5% $CO_2$. Once cells reached 80–90% confluency, they were washed once with EDTA solution and incubated in accutase (Sigma-Aldrich) for 2 min at 37°C. Cells were resuspended in ice-cold FACS buffer (2% FBS in PBS) and transferred to a round-bottomed 96-well plate at ~10⁶ cells/well. Cells were washed once, and cell suspensions were centrifuged at 300 g for 5 min. The supernatant was removed, and cells were resuspended in FACS buffer. Cells were incubated with a panel of seven fluorescein-labeled lectins (final concentration 20 µg/ml; Vector Biolabs) and a fixable viability dye, eFluor 780 (1:1,000, Thermo Fisher Scientific), diluted in FACS buffer on ice in darkness for 30 min. Where specified, controls to validate lectin specificities were included, and lectins were preincubated in FACS buffer containing saturating concentrations of competitive sugars at least 30 min before addition to cells. Cells were washed 3 times in FACS buffer and fixed in 4% PFA in PBS for 20 min at room temperature. Cells were washed a further two times in FACS buffer and kept at 4°C in darkness until required. Cell suspensions were filtered using a 100-µm plate filter immediately before analysis on an LSRII flow cytometer (BD Biosciences) or an EC800 flow cytometer (Sony). Data were analyzed and histogram plots generated using FlowJo v10. Singlets were gated according to forward and side scatter profiles, and dead cells were excluded from analysis using the viability stain. Single-color control samples were included to confirm the appropriate compensation parameters.

## Flow cytometry assay for Golgi retention

Inducible stable cell lines expressing GFP-tagged Golgi enzyme chimeric reporters were cultured in six-well plate format in selection medium containing 60 µg/ml cumate for at least a week before analysis. Once cells reached 80–90% confluency, they were washed once with EDTA solution and incubated in accutase for 2 min at 37°C. Cells were resuspended in selection medium and transferred into a deep 96-well plate. Cells were washed once by centrifugation at 300 g for 5 min, followed by resuspension in ice-cold FACS buffer. Cells were transferred to a round-bottomed 96-well plate and incubated on ice in darkness for 30 min with an AF647-conjugated anti-GFP antibody (1:20, BioLegend) and fixable viability dye eFluor 780 (1:1,000, BD Biosciences) diluted in FACS buffer. Cells were subsequently washed, fixed, and analyzed as described for lectin stains. Furthermore, GFP-negative cells were excluded from analysis, and a ratio of the AF647 signal to the GFP signal was used to derive a quantitative parameter for Golgi retention.

## Immunofluorescence

Cells were seeded onto multispot microscope slides (Hendley-Essex) in culture medium in a humidified incubator at 37°C with 5% $CO_2$. 24 h after seeding, cells were washed twice in PBS and fixed in 4% PFA in PBS for 20 min at room temperature. Cells were washed twice in PBS and permeabilized for 10 min in 10% Triton X-100. Cells were washed five times in PBS and blocked for 1 h in blocking buffer (20% FBS, 1% Tween-20 in PBS). Blocking buffer was aspirated, and cells were incubated in the primary antibody cocktail diluted in blocking buffer for 1 h. Cells were washed twice in PBS, incubated in blocking buffer for 10 min, and washed twice again in PBS before incubation for 1 h in blocking buffer containing fluorescent secondary antibodies (Data S5). Cells were washed twice in PBS, incubated in blocking buffer for 10 min, and washed twice again. PBS was aspirated, and Vectashield mounting medium (Vector Biolabs) was added to cells before application of the coverslip. The coverslip was sealed with nail varnish, and slides were imaged at room temperature using a 63×/1.4 oil-immersion objective on a Leica TCS SP8 confocal microscope controlled with Leica Application Suite X. Images were normalized without altering gamma using Adobe Photoshop CC 2017.

## Mass spectrometry
### Protein digestion

Protein samples (10 × 200 µg each) in lysis buffer (8 M urea and 20 mM Tris, pH 8) were reduced with 5 mM DTT at 56°C for 30 min and alkylated with 10 mM iodoacetamide in the dark at room temperature for 30 min. The samples were then diluted to 4 M urea and digested with Lys-C (Promega), 67:1 (protein/Lys-C ratio, wt/wt) for 4 h at 25°C. Next, the samples were further diluted to 1.6 M urea and digested with trypsin (Promega) 50:1 (protein/trypsin, wt/wt) overnight at 25°C. Digestion was stopped by the addition of formic acid (FA) to a final concentration of 0.5%. Any precipitates were removed by centrifugation at 13,000 rpm for 8 min. The supernatants were desalted using homemade C18 stage tips (3M Empore) contained 4 mg of Poros R3 (Applied Biosystems) resin. Bound peptides were eluted with 30–80% acetonitrile (MeCN) in 0.1% TFA and lyophilized.

### TMT labeling

Peptide mixtures from each condition were resuspended in 74 µl of 200 mM Hepes, pH 8.3. 36 µl (720 µg) TMT 10plex reagent

(Thermo Fisher Scientific) was added and incubated at room temperature for 1 h. The labeling reaction was then terminated by incubation with 7.3 µl of 5% hydroxylamine. The labeled peptides were pooled into a single sample and desalted using the stage tips method above.

### Offline, high-pH, reverse-phase peptide fractionation
Approximately 200 µg of the labeled peptides were separated on an offline HPLC. The experiment was performed using XBridge BEH130 C18, 5 µm, 2.1 × 150 mm (Waters) column with XBridge BEH C18 5 µm Van Guard cartridge, connected to an Ultimate 3000 Nano/Capillary LC System (Dionex). Peptides were separated with a gradient of 1–90% B (A: 5% MeCN/10 mM ammonium bicarbonate, pH 8; B: MeCN/10 mM ammonium bicarbonate, pH 8, [9:1]) for 60 min at a flow rate of 250 µl/min. A total of 54 fractions were collected, combined into 18 fractions, and lyophilized. Dried peptides were resuspended in 1% MeCN/0.5% FA and desalted using stage tips for mass spectrometry analysis.

### Mass spectrometry analysis
Peptides were separated on an Ultimate 3000 RSLC nano System (Thermo Fisher Scientific), using a binary gradient consisting of buffer A (2% MeCN and 0.1% FA) and buffer B (80% MeCN and 0.1% FA). Eluted peptides were introduced directly via a nanospray ion source into a Q Exactive Plus hybrid quadrupole-Orbitrap mass spectrometer (Thermo Fisher Scientific). The mass spectrometer was operated in standard data-dependent mode, performed survey full-scan (MS, m/z = 380–1,600) with a resolution of 70,000, followed by MS2 acquisitions of the 15 most intense ions with a resolution of 35,000 and normalized collision energy of 33%. MS target values of 3e6 and MS2 target values of 1e5 were used. Dynamic exclusion was enabled for 40 s.

### GST affinity chromatography mass spectrometry
Gel samples were destained with 50% (vol/vol) acetonitrile and 50 mM ammonium bicarbonate, reduced with 10 mM DTT, and alkylated with 55 mM iodoacetamide. Digestion was with 6 ng/µl trypsin (Promega) overnight at 37°C, and peptides were extracted in 2% (vol/vol) FA and 2% (vol/vol) acetonitrile and analyzed by nano-scale capillary LC-MS/MS (Ultimate U3000 HPLC, Thermo Fisher Scientific, Dionex) at a flow of ∼300 nL/min. A C18 Acclaim PepMap100 5 µm, 100 µm × 20 mm nanoViper (Thermo Fisher Scientific, Dionex), trapped the peptides before separation on PicoChip column: 75 µm internal diameter × 15 µm tip packed with 105 mm 3 µm Reprosil-PUR C18-AQ 120A (New Objective). Peptides were eluted with an acetonitrile gradient. The analytical column outlet was interfaced via a nanoflow electrospray ionization source with a linear ion trap mass spectrometer (Orbitrap Velos, Thermo Fisher Scientific). Data-dependent analysis was performed using a resolution of 30,000 for the full MS spectrum, followed by 10 MS/MS spectra in the linear ion trap. MS spectra were collected over a m/z range of 300–2,000. MS/MS scans were collected using a threshold energy of 35 for collision-induced dissociation. LC-MS/MS data were searched against the UniProt KB database using Mascot (Matrix Science), with a precursor tolerance of 10 ppm and a fragment ion mass tolerance of 0.8 D. Two missed enzyme cleavages and variable modifications for oxidized methionine, carbamidomethyl cysteine, pyroglutamic acid, phosphorylated serine, threonine, and tyrosine were included. MS/MS data were validated using the Scaffold program (Proteome Software).

### Mass spectrometry data analysis
The acquired MS/MS raw files were processed using MaxQuant (Cox and Mann, 2008), with the integrated Andromeda search engine (v1.6.6.0). MSMS spectra were searched against *Homo sapiens* UniProt Fasta database. Carbamidomethylation of cysteines was set as fixed modification, while methionine oxidation and N-terminal acetylation (protein) were set as variable modifications. Protein quantification requires 1 (unique + razor) peptide. Other parameters in MaxQuant were set to default values. MaxQuant output file, proteinGroups.TXT was then processed with Perseus software (v1.6.6.0). After uploading the matrix, the data were filtered to remove identifications from reverse database, modified peptide only, and common contaminants.

### Bioinformatics
Type II TM proteins and their TM span locations were initially identified from the reviewed, nonredundant UniProt entries for the human proteome (UniProt Consortium, 2021). Entries were manually reviewed to correct obvious errors, which mostly related to subcellular localization and signal peptide annotation. TM span edges were then refined using a single, consistent approach employed previously (Parsons et al., 2019). In précis, this considered positions ±5 residues from the stated UniProt TM edge, found the point of maximum hydrophobicity difference between the five preceding and five subsequent residues, and then trimmed the end residue if hydrophilic (here using Arg, Lys, Asp, Glu, Gln, Asn, His, or Ser) or extended if a hydrophobic residue was next (Phe, Met, Ile, Leu, Val, Cys, Trp, Ala, Thr, or Gly). Subsequently, the TM protein entries were matched to the protein IDs used in various analysis groups via the gene name to accession code mapping (gene2acc) at UniProt, which ties redundant protein entries to their gene of origin.

Logo plots were generated using the Python script available at https://github.com/tjs23/logo_plot/. Inputs to the plots were one-letter protein sequences of TM spans, aligned on their first residue, with flanking regions. These regions covered positions from 15 residues before to 35 residues after each TM start position (the N-terminal edge), which also acted as the anchor point to compare the TM spans from different proteins. Where TM spans had short flanking tails that did not reach the edge of the plot regions, the ends of the protein sequences were padded with X, which was plotted with the real amino acid types. This proved helpful to illustrate the occurrence of short tails.

From the curated set of type II TM proteins, those with a known, unambiguous subcellular localization within the membranes of the ER, Golgi apparatus, or plasma membrane were selected for analysis of positively charged near-TM groups. These groups included arginine side chains, lysine side chains, or an N-terminal α-amino group. Counts were made for the occurrence of these within a six-residue region just outside the TM span from the cytoplasmic TM edge, as described above.

## Online supplemental material

Fig. S1 shows validation of the *ΔGOLPH3;ΔGOLPH3L* U2OS cell line. Fig. S2 shows flow cytometry gating strategies. Fig. S3 shows genetic rescue of the destabilization of Golgi residents in *ΔGOLPH3;ΔGOLPH3L* cells. Fig. S4 shows that GOLPH3 interacts with the tails of GALNT2 and GALNT7. Data S1 lists GST affinity chromatography data for GOLPH3 and GOLPH3L. Data S2 compares affinity chromatography data to COPI vesicle proteome. Data S3 shows proteomic data comparing WT and *ΔGOLPH3;ΔGOLPH3L* cells. Data S4 lists proteins used for bioinformatic analyses. Data S5 lists antibodies, plasmids, and oligonucleotides.

## Acknowledgments

We thank David Owen, Jonathan Kaufman and Jasmine Cornish for discussions on the structure of GOLPH3. Thanks to Ryan Britnell for help with analysis of the proteomics data and to Maria Daly and Fan Zhang for help with cell sorting and flow cytometry advice. GOLPH3 cDNA was a kind gift from David Gershlick.

Funding was from the Medical Research Council, as part of UK Research and Innovation, file reference number MC_U105178783.

The authors declare no competing financial interests.

Author contributions: Conceptualization: L.G. Welch and S. Munro. Formal analysis: L.G. Welch. Funding acquisition and supervision: S. Munro. Investigation and methodology: L.G. Welch, S-Y. Peak-Chew, and F. Begum. Visualization: L.G. Welch and T.J. Stevens. Writing – original draft: L.G. Welch. Writing – review and editing: S. Munro.

Submitted: 18 June 2021

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

## Supplemental material

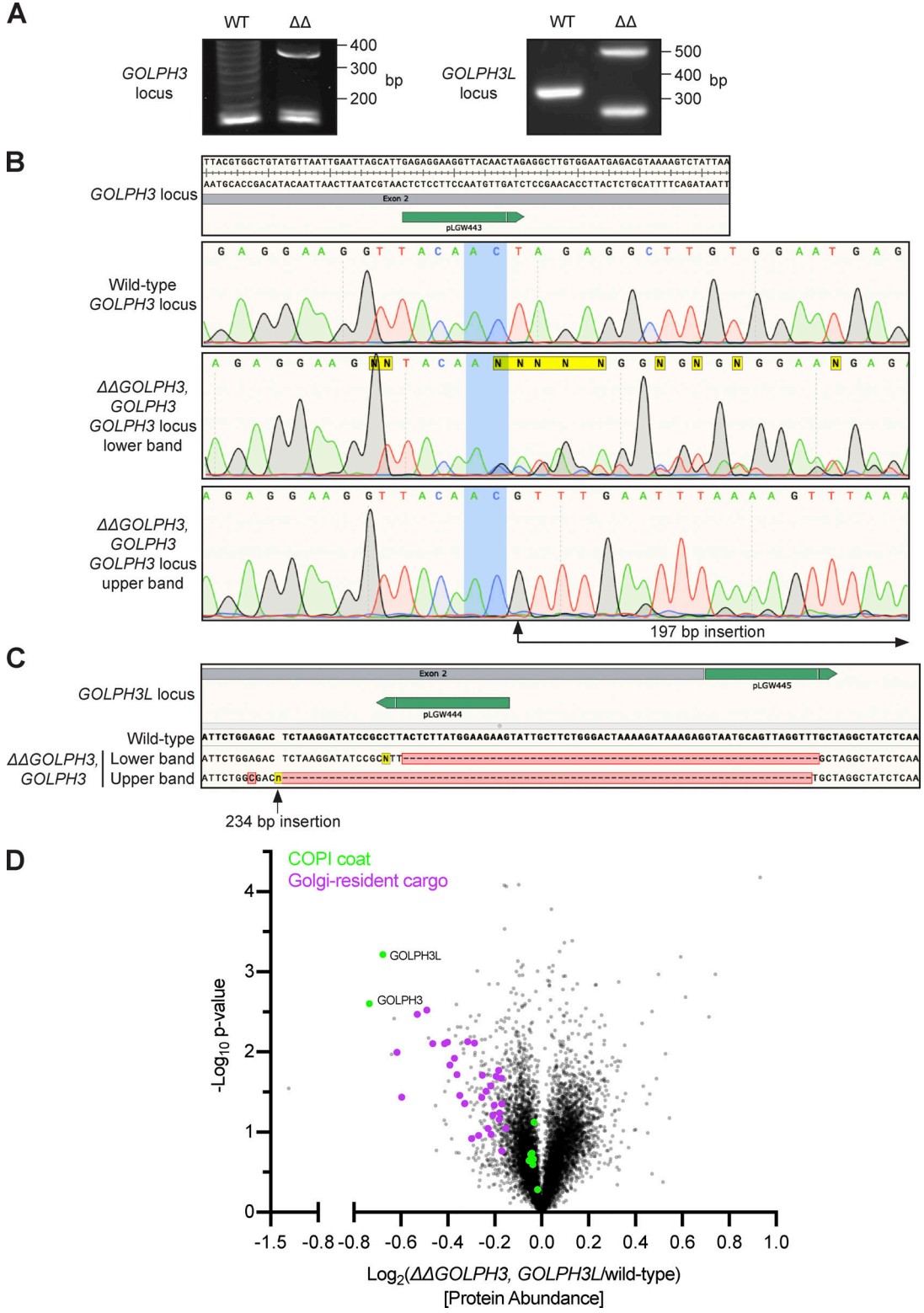

Figure S1. **Validation of the ΔGOLPH3;ΔGOLPH3L U2OS cell line. (A)** Agarose gels resolving PCR-amplified regions of the genomic loci of *GOLPH3* and *GOLPH3L* targeted by CRISPR-Cas9 gene editing. **(B)** *GOLPH3* was targeted at one site in exon 2 (plasmid pLGW443 encoding guide 5′-GAGAGGAAGGTTACAACT AG-3′ [green line]), inducing small indel mutations in at least two alleles (lower band) and a 197-bp out-of-frame insertion in another allele (upper band). **(C)** *GOLPH3L* was targeted at a site in exon 2 and a site in intron 2–3 (plasmids pLGW444 for guide 5′-CTTCTTCCATAAGAGTAAGG-3′ and pLGW445 for 5′-GTA ATGCAGTTAGGTTTGCT-3′), inducing a 62-bp deletion in one allele and a 79-bp deletion with a 234-bp insertion in the other allele. **(D)** A volcano plot comparing spectral intensity values corresponding to the individual proteins in ΔGOLPH3;ΔGOLPH3L cells versus WT U2OS cells. The datasets were generated from a duplicate of repeats and were Z-score normalized according to the median; P values were generated with a Student's *t* test. Points correspond to individual proteins. Notable proteins displaying a large difference are colored: GOLPH3 proteins and COPI (green), Golgi-resident cargo (magenta).

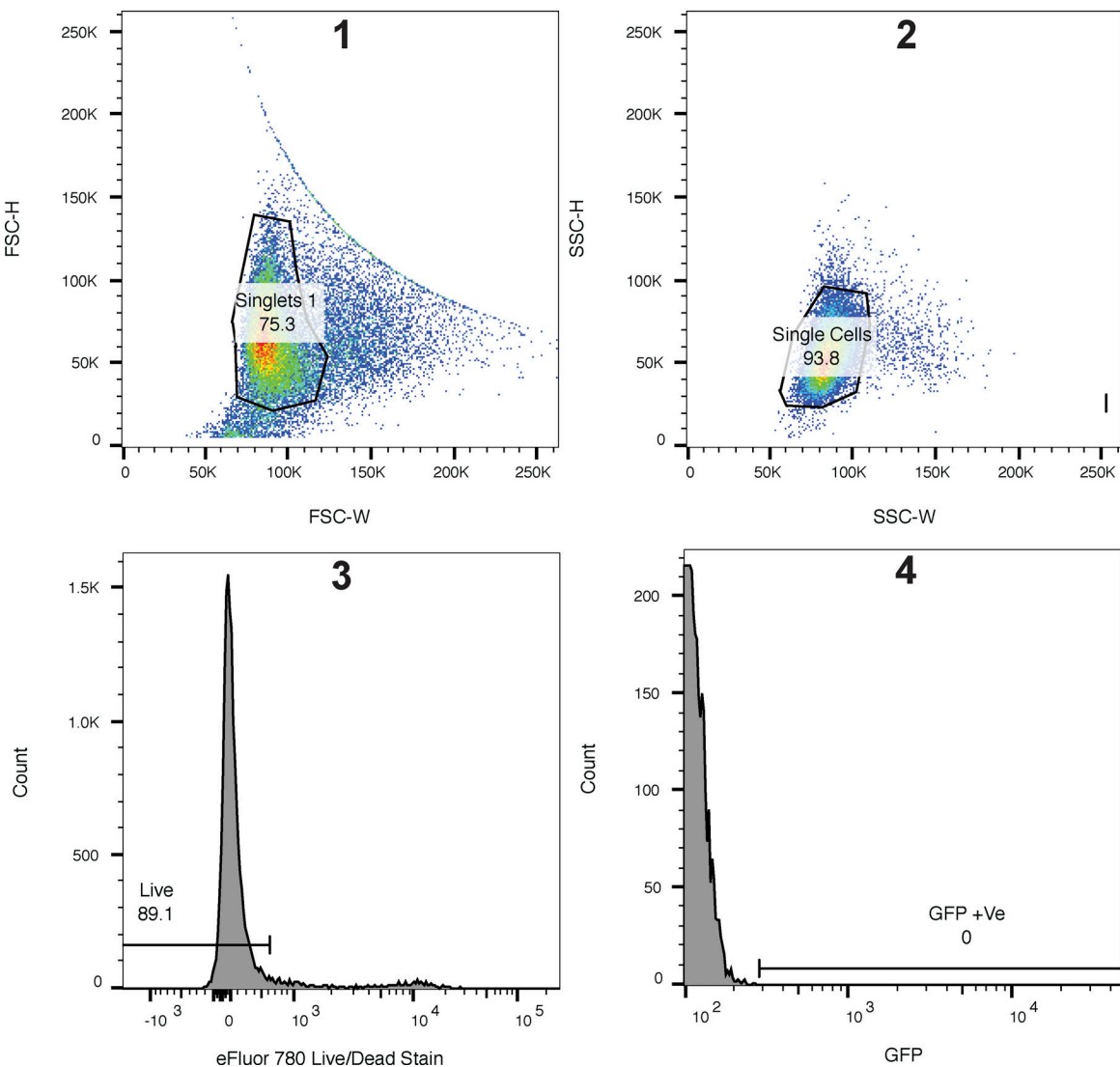

Figure S2. **Flow cytometry gating strategies.** A representative gating strategy for the in vivo Golgi retention assays and the lectin staining. WT stained U2OS cells were used to assign the gates. Plots represent 25,000 events. Hierarchical gating strategy in order from 1 to 4: top panels show isolation of singlets based on forward (FSC) and side (SSC) scatter using height (H) and width (W). Lower panels show gating for live cells using an eFluor 780 fixable live/dead stain followed by a gate for GFP-positive cells (or FITC-positive cells for lectin stains). Compensation was done using single color controls on an LSRII, and plots and gates were generated using FlowJo v10.

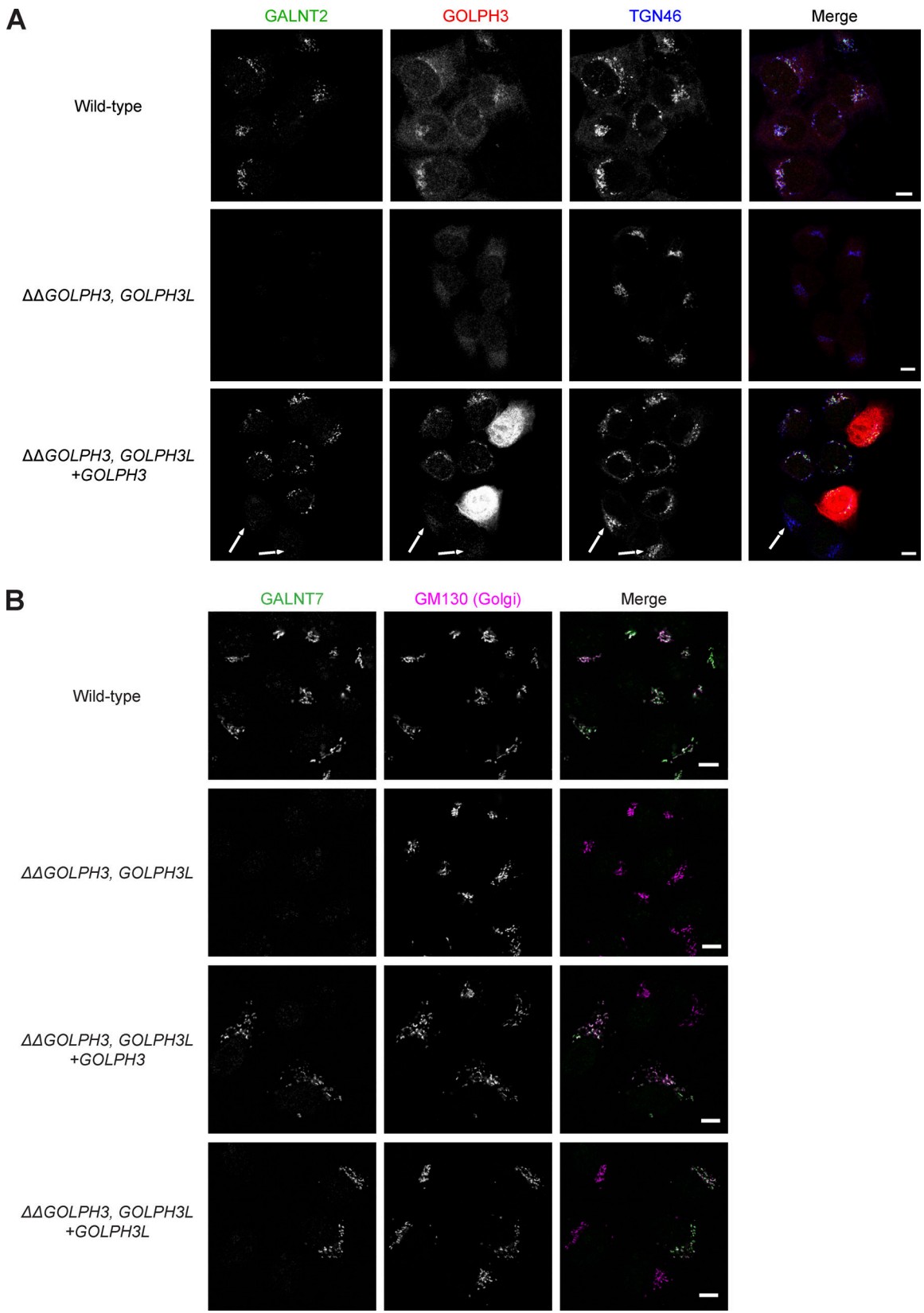

Figure S3. **Genetic rescue of the destabilization of Golgi residents in ΔΔGOLPH3;ΔGOLPH3L cells. (A)** Confocal micrographs of WT and *ΔGOLPH3; ΔGOLPH3L* U2OS cells or a polyclonal rescue cell line stably reexpressing *GOLPH3* in the *ΔGOLPH3;ΔGOLPH3L* background. Arrows indicate cells that do not express detectable levels of GOLPH3 and so lack the rescue of the loss of the Golgi resident GALNT2. TGN46 (Golgi/TGN marker). **(B)** As in A but WT or *ΔGOLPH3;ΔGOLPH3L* U2OS cell lines transiently transfected with plasmids encoding GOLPH3 or GOLPH3L as indicated, and labeled for GALNT7 (Golgi resident enzyme) and GM130 (Golgi marker). Scale bars, 10 µm.

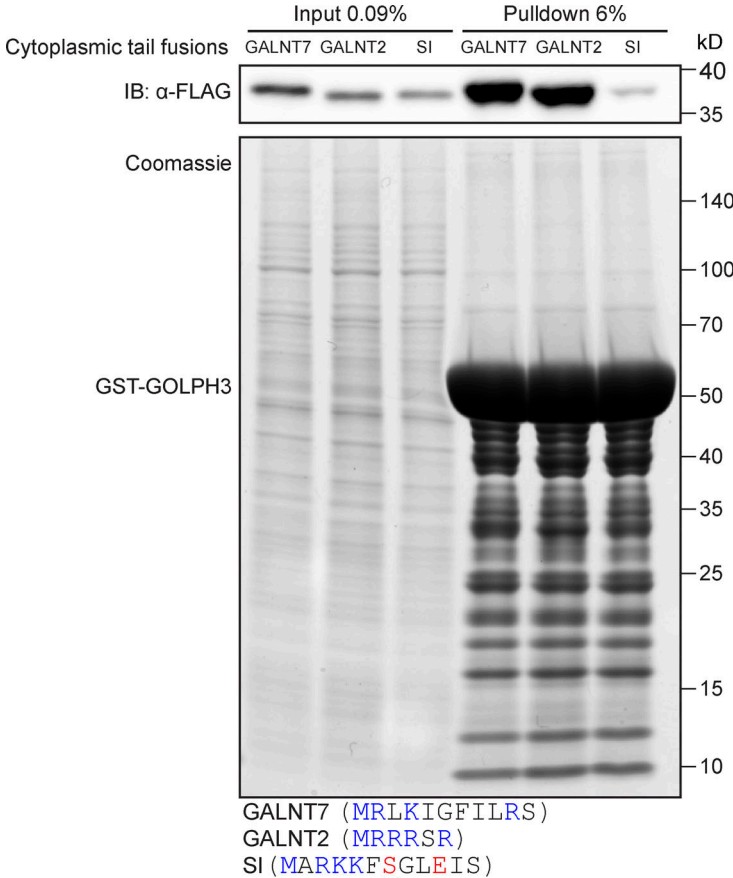

GALNT7 (MRLKIGFILRS)
GALNT2 (MRRRSR)
SI (MARKKFSGLEIS)

Figure S4. **GOLPH3 interacts with the tails of GALNT2 and GALNT7.** Test of the ability of bacterially expressed GST-tagged GOLPH3 to pull down different GFP-FLAG-tagged cytoplasmic tail chimeras from 293T cell lysate. Experiment representative of two independent replicates. Sequences of the cytoplasmic tails are given below; charged residues are colored blue (positive) or red (negative). IB, immunoblotting. Note that the tail of SI is predicted to be phosphorylated on the serine at position 7.

**Provided online are five datasets. Data S1 lists GST affinity chromatography data for GOLPH3 and GOLPH3L. Data S2 compares affinity chromatography data to COPI vesicle proteome. Data S3 shows proteomic data comparing WT and ΔGOLPH3;ΔGOLPH3L cells. Data S4 lists proteins used for bioinformatic analyses. Data S5 lists antibodies, plasmids, and oligonucleotides.**

