## [Peer Review File · The Journal of Cell Biology]

GOLPH3 and GOLPH3L are broad-spectrum COPI adaptors for sorting into intra-Golgi transport vesicles

Lawrence Welch, Sew-Yeu Peak-Chew, Farida Begum, Tim Stevens, and Sean Munro

Corresponding Author(s): Sean Munro, MRC Laboratory of Molecular Biology

Review Timeline:

Submission Date:	2021-06-18
Editorial Decision:	2021-07-28
Revision Received:	2021-08-08

Monitoring Editor: Ira Mellman

Scientific Editor: Andrea Marat

Transaction Report:

DOI: <https://doi.org/10.1083/jcb.202106115>

July 28, 2021

RE: JCB Manuscript #202106115

Dr. Sean Munro
MRC Laboratory of Molecular Biology
Francis Crick Avenue
Cambridge CB2 0QH
United Kingdom

Dear Sean:

Thank you for submitting your manuscript entitled "GOLPH3 and GOLPH3L are broad-spectrum COPI adaptors for sorting into intra-Golgi transport vesicles" to JCB. As you will see, the three expert reviewers are all very positive regarding the quality and impact of your work. Therefore, we would be happy to publish your paper in JCB pending final revisions necessary to meet our formatting guidelines (see details below). In your final revision, please be sure to address the reviewer comments with appropriate edits to as well as an expanded discussion in your text.

A. MANUSCRIPT ORGANIZATION AND FORMATTING:

Full guidelines are available on our Instructions for Authors page, <https://jcb.rupress.org/submission-guidelines#revised>. **Submission of a paper that does not conform to JCB guidelines will delay the acceptance of your manuscript.**

1) Text limits: Character count for Articles is < 40,000, not including spaces. Count includes title page, abstract, introduction, results, discussion, acknowledgments, and figure legends. Count does not include materials and methods, references, tables, or supplemental legends.

2) Figures limits: Articles may have up to 10 main text figures.

3) Figure formatting: Scale bars must be present on all microscopy images, including inset magnifications. Molecular weight or nucleic acid size markers must be included on all gel electrophoresis.

4) Statistical analysis: Error bars on graphic representations of numerical data must be clearly described in the figure legend. The number of independent data points (n) represented in a graph must be indicated in the legend. Statistical methods should be explained in full in the materials and methods. For figures presenting pooled data the statistical measure should be defined in the figure legends. Please also be sure to indicate the statistical tests used in each of your experiments (either in the figure legend itself or in a separate methods section) as well as the parameters of the test (for example, if you ran a t-test, please indicate if it was one- or two-sided, etc.). Also, if you used parametric tests, please indicate if the data distribution was tested for normality (and if so, how). If not, you must state something to the effect that "Data distribution was assumed to be

normal but this was not formally tested."

5) Abstract and title: The abstract should be no longer than 160 words and should communicate the significance of the paper for a general audience. The title should be less than 100 characters including spaces. Make the title concise but accessible to a general readership.

6) Materials and methods: Should be comprehensive and not simply reference a previous publication for details on how an experiment was performed. Please provide full descriptions in the text for readers who may not have access to referenced manuscripts.

7) Please be sure to provide the sequences for all of your primers/oligos and RNAi constructs in the materials and methods. You must also indicate in the methods the source, species, and catalog numbers (where appropriate) for all of your antibodies. Please also indicate the acquisition and quantification methods for immunoblotting/western blots.

8) Microscope image acquisition: The following information must be provided about the acquisition and processing of images:

a. Make and model of microscope

b. Type, magnification, and numerical aperture of the objective lenses

c. Temperature

d. Imaging medium

e. Fluorochromes

f. Camera make and model

g. Acquisition software

h. Any software used for image processing subsequent to data acquisition. Please include details and types of operations involved (e.g., type of deconvolution, 3D reconstitutions, surface or volume rendering, gamma adjustments, etc.).

10) Supplemental materials: There are strict limits on the allowable amount of supplemental data. Articles may have up to 5 supplemental display items (figures and tables). Please also note that tables, like figures, should be provided as individual, editable files. A summary of all supplemental material should appear at the end of the Materials and methods section.

13) ORCID IDs: ORCID IDs are unique identifiers allowing researchers to create a record of their various scholarly contributions in a single place. At resubmission of your final files, please consider providing an ORCID ID for as many contributing authors as possible.

B. FINAL FILES:

Thank you for this interesting contribution, we look forward to publishing your paper in Journal of Cell Biology.

Sincerely,

Ira Mellman, Ph.D.
Editor

Andrea L. Marat, Ph.D.
Senior Scientific Editor

Journal of Cell Biology

Reviewer #1 (Comments to the Authors (Required)):

This paper describes the analysis of the Golgi protein GOLPH3 and its closely related paralogue GOLPH3L. These proteins have been implicated in a number of functions at the Golgi apparatus, including the recycling of Golgi-resident enzymes in COPI vesicles to maintain their retention at this organelle. Here, the authors use CRISPR-mediated knockout, combined with biochemistry and an innovative Golgi retention assay to show that the GOLPH3 proteins function as COPI adaptors for numerous Golgi enzymes. The systematic analysis is very convincing and shows the key role of the GOLPH3 protein at the Golgi apparatus. Indeed, the authors convincingly show their loss results in gross changes in protein glycosylation. Detailed analysis of the Golgi residents themselves is performed to identify the nature of the retention signal bound by the GOLPH3 proteins, corresponding to a cluster of basic residues within a short cytoplasmic tail.

The paper is of very high quality, and the data are very convincing. The results represent an important advance in the field that will be of much interest to the cell biology community and beyond, especially considering that GOLPH3 is an oncogene. I am very supportive of publication.

My only suggestion relates to individual knockouts of GOLPH3 and GOLPH3L. Unless I missed it, this is not mentioned at all. I feel it should be commented on in the results or discussion. Were individual knockouts created, and what was the phenotype, if any? Are the proteins functionally redundant?

Reviewer #2 (Comments to the Authors (Required)):

This manuscript by Welch et al describes a study of the roles of GOLPH3 and GOLPH3L proteins at the Golgi apparatus. The study builds on published research using a variety of organisms that established that GOLPH3 directly binds to the N-terminal cytoplasmic type II Golgi of a subset of Golgi residents, chiefly glycosylation enzymes to effect their retention within the Golgi apparatus (and to a lesser extent GOLPH3L). GOLPH3 and yeast Vps74 were also previously shown to associate with COPI coat subunits. In this study, Welch et al sought to identify the full repertoire of GOLPH3/GOLPH3L ('GOLPH3/L') clients to more fully elucidate their roles in the Golgi. They identify new proteins that associate with GOLPH3/L and that are destabilized due to a failure to be retained within the Golgi in cells lacking GOLPH3/L using a variety of sensitive proteomics-based approaches. Bioinformatics analyses, in vitro binding assays and cell-based assays of Golgi retention identified positively charged residues that are enriched in the cytoplasmic portions of GOLPH3/L clients and identified a patch on GOLPH3/L that appears to recognize this portion of the clients. The data prompt a re-evaluation of the proposed roles of GOLPH3/L in secretion and the argument (Rizzo et al) that sphingolipid biosynthetic enzymes are the predominant GOLPH3/L clients.

Throughout, the experiments are conducted with a high degree of rigor and I have no technical concerns to raise. Though the results do not break new conceptual ground, overall, the study more firmly establishes GOLPH3/L as a broad specificity COPI adapter that is key to maintaining the glycosylation capacity (and likely other proteins) of the Golgi apparatus. In this respect the study is an important advance in understanding the organization of the Golgi apparatus. The clarity and scholarship of the manuscript superb. I recommend publication without delay.

There is a typo on page 6: "(42 common hits, 13 GOLPH3-specific and 18 GOLPH3-specific)" should read "...18 GOLPH3L-specific".

Reviewed by Chris Burd

Reviewer #3 (Comments to the Authors (Required)):

There is considerable excitement about the peripheral membrane protein GOLPH3 and its paralogue GOLPH3L, which have been identified as adaptors for capturing glycosylation enzymes into COPI vesicles at the Golgi. However, other functions have been proposed for GOLPH3+3L, and the mechanism and extent of their role in transporting Golgi enzymes are incompletely understood. This project employs proteomics, binding assays, bioinformatics, and a live cell assay for Golgi retention to characterize the clients of GOLPH3+3L and their mode of interaction with Golgi enzymes. GOLPH3+3L are shown to interact with membrane-proximal basic residues in a diverse group of Golgi enzyme cytosolic tails.

This analysis is very thorough. It is also elegantly presented and well controlled. The findings add substantially to our understanding of GOLPH3+3L, and they will be of interest to a broad range of investigators who work on the properties of the Golgi or on glycosylation. I have only a few comments.

[1] There may be a fly in the ointment: according to tomography data, COPI vesicles are largely absent from the TGN, which is the site of PtdIns4P synthesis. Is it clear that the PtdIns4P-dependent GOLPH3+3L proteins are actually present at the right time and place to act as COPI adaptors?

[2] For the GALNT2 chimera, Figure 2C (fluorescence microscopy) and Figure 3B (flow cytometry) do not align. According to Figure 2C, much of the chimera remains intracellular in the double deletion, but according to Figure 3B, the chimera behaves like a cell surface protein in the double deletion. Is it possible that antibody binding in the flow cytometry system interferes with a dynamic cycle of endocytosis and cell surface delivery?

[3] Figure 6A was confusing at first. Maybe the edge of the TM region (position 15) could be marked at the top together with a left-pointing arrow labeled "cytosolic tail".

Revisions in Response to Reviewers' Comments.

We are very grateful indeed to the reviewers for their positive comments about our work and their constructive suggestions for improvements. We have followed these suggestions as described below, and have also made some minor changes to ensure that the manuscript conforms to the JCB guidelines.

Reviewer #1 (Comments to the Authors):

My only suggestion relates to individual knockouts of GOLPH3 and GOLPH3L. Unless I missed it, this is not mentioned at all. I feel it should be commented on in the results or discussion. Were individual knockouts created, and what was the phenotype, if any? Are the proteins functionally redundant?

GOLPH3 and GOLPH3L are expressed a very different levels, and so to allow us to directly test and compare their functions our strategy was to remove both GOLPH3 and GOLPH3L, and to then transfect them back individually to test their roles. Previous studies have investigated the individual knockouts of GOLPH3 and GOLPH3L, and this is mentioned in the manuscript. We agree that this point should have been properly discussed, and we have now added a comment on this near the start of the Discussion, as follows:

"Previous studies have shown that removal of GOLPH3 alone from mammalian cell lines results in defects in the retention of particular enzymes, but GOLPH3L is present at only ~10% of the level of GOLPH3 in cultured cell lines, and mRNA sequencing analysis indicates a similarly lower expression across most tissues (Consortium, 2020; Bekker-Jensen et al., 2017). There have been no reported investigations of its function, apart from a suggestion that it is a negative regulator of GOLPH3 function (Ng et al., 2013). Our in vitro binding and in vivo over-expression studies suggest that it has similar properties to GOLPH3, and so it may make a major contribution to retention in some cell types. However, further work will be needed to resolve its precise role, and our inability to isolate cell lines stably over-expressing the protein does at least suggest some potential negative effect at high levels."

Reviewer #2 (Comments to the Authors):

There is a typo on page 6: "(42 common hits, 13 GOLPH3-specific and 18 GOLPH3-specific)" should read "...18 GOLPH3L-specific".

This typo has been corrected.

Reviewer #3 (Comments to the Authors):

[1] There may be a fly in the ointment: according to tomography data, COPI vesicles are largely absent from the TGN, which is the site of PtdIns4P synthesis. Is it clear that the PtdIns4P-dependent GOLPH3+3L proteins are actually present at the right time and place to act as COPI adaptors?

We apologise for not discussing PI4P and the localisation of GOLPH3+3L in more detail. The relevant data is not extensive, and indeed the best localisation data is from yeast rather than mammals, nonetheless it is potentially an interesting and important issue. We have thus added a section to the Discussion as follows:

"The COPI coat not only forms the vesicles that bud from Golgi compartments to mediate intra-Golgi traffic, but it also forms the vesicles that capture escaped ER residents in the earliest Golgi compartments to recycle them back to the ER (Bykov et al., 2017; Adolf et al., 2019; Cosson and Letourneur, 1994). Immuno-EM suggests that GOLPH3 is found on the latter half of the Golgi stack, and imaging in yeast indicates that Vps74 is more abundant in the cis and medial cisternae than in the TGN (Wood et al., 2012; Wu et al., 2000). In both yeast and mammals, Golgi recruitment of Vps74/GOLPH3 depends on PtdIns4P, which is relatively enriched at the TGN but apparently present at lower levels on earlier compartments (Graham and Burd, 2011; Wood et al., 2009; Dippold et al., 2009). Indeed, it has been proposed that in yeast the Sac1 PtdIns4P-phosphatase acts in earlier compartments to degrade the PtdIns4P that recycles from later compartments and hence allows the release of Vps74 from membranes, and in mammals, disruption of Sac1 activity has been reported perturbs the localisation of particular Golgi residents (Wood et al., 2012; Cheong et al., 2010). Further

work will be needed to determine the precise distribution of GOLPH3, COPI and PtdIns4P within the stack, but the dependence on PtdIns4P could allow GOLPH3 to be selectively recruited into the COPI vesicles that form on later cisternae. This would allow COPI vesicles to form at the earliest Golgi compartments without GOLPH3 and hence recycle escaped ER residents rather than Golgi proteins, whilst the COPI vesicles that form on later cisternae that contain more PI4P would be equipped with GOLPH3 to mediate intra-Golgi recycling of the Golgi residents."

[2] For the GALNT2 chimera, Figure 2C (fluorescence microscopy) and Figure 3B (flow cytometry) do not align. According to Figure 2C, much of the chimera remains intracellular in the double deletion, but according to Figure 3B, the chimera behaves like a cell surface protein in the double deletion. Is it possible that antibody binding in the flow cytometry system interferes with a dynamic cycle of endocytosis and cell surface delivery?

The antibody binding for the flow cytometry assay is done at 4°C (as described in the Methods), under which conditions there is no endocytosis or cell surface delivery and hence nothing to interfere with. Instead, the appearance of the chimera in Figure 2C reflects the fact that protein in the Golgi is concentrated in a small area and so easy to see, whereas the cell surface pool is spread out over a much larger area, and also the plasma membrane is harder to image as it is difficult to get the entire cell in the c 1 micron focal plane of the confocal microscope. Indeed, even the PM reporter shows a relatively bright Golgi in Figure 2C. It was for these reasons that we developed the more quantitative flow cytometry assay which measures the entire cell, and also obtains data from thousands of cells. We have amended the description of the assay in the Results to include the fact the antibody binding is at 4°C, and added a note on the immunofluorescence to the legend of Figure 2C as follows:

"The small area of the Golgi makes the intracellular population of reporter clearly visible, and although all the reporters are visible on the plasma membrane in the knockout, the large area and the height of the cell make the surface levels of reporter harder to accurately assess, hence our development of a flow cytometry based quantitative assay for surface expression."

[3] Figure 6A was confusing at first. Maybe the edge of the TM region (position 15) could be marked at the top together with a left-pointing arrow labeled "cytosolic tail".

This is a very helpful suggestion, and we have now labelled the edge of the TM region with a dotted line, and this is explained in the figure legend.